# Relationships between Spatial and Temporal Variations in Precipitation, Climatic Indices, and the Normalized Differential Vegetation Index in the Upper and Middle Reaches of the Heihe River Basin, Northwest China

**Fanglei Zhong** [1], **Qingping Cheng** [2,3,*] **and Yinchun Ge** [2]

1   School of Economics, Lanzhou University, Lanzhou 730000, China
2   Northwest Institute of Eco-Environmental and Resources, Chinese Academy of Sciences, Lanzhou 730000, China
3   University of Chinese Academy of Sciences, Beijing 100049, China
*   Correspondence: qpchengtyli@foxmail.com; Tel.: +86-178-0596-3632

**Abstract:** Changes in precipitation are critical indicators of climate change. In this study, the daily precipitation records from 10 meteorological stations in the Heihe River Basin, Northwest China from 1961–2016, precipitation indices, climate indices, and the normalized differential vegetation index (NDVI) were investigated using the Pearson, Kendall, and Spearman correlation coefficients; Theil-Sen Median; Mann–Kendall test; and wavelet coherence. The results indicated that the occurrences (fractional contributions) of 1–2-day wet periods were 81.3% (93.9%) and 55.3% (82.1%) in the upper (UHRB) and middle (MHRB) reaches of the Heihe River Basin, respectively. The spatial distribution of the occurrences (fractional contributions) was almost consistent with non-significant increases/decreases at stations. The ATP, ATD, API, and AMRD increased, while precipitation regimes suggest that dry seasons are getting wetter, and wet seasons are getting drier, although these changes were not significant. Wavelet coherence analyses showed that climate indices influenced precipitation, mainly its concentration, on a 4- to 78.6-month timescale. The Pearson, Kendall, and Spearman correlation coefficients showed weak lagged linkages between precipitation and the North Arctic Oscillation (NAO), Pacific Decadal Oscillation (PDO), and Atlantic Multidecadal Oscillation (AMO). The NDVI of grasslands, meadows and coniferous forests was significantly and positively correlated with precipitation, while the NDVI of alpine vegetation, swamps and shrubs was negatively and significantly correlated with precipitation in the UHRB. The NDVI of grasslands was significantly and positively correlated, but the NDVI of shrubs, coniferous forests and cultivated vegetation was negatively and significantly correlated with precipitation in the MHRB. The correlation between cultivated vegetation and natural precipitation in the MHRB may have been weakened by human activities.

**Keywords:** precipitation indices; large-scale circulations; NDVI; Heihe River Basin

## 1. Introduction

It is recognized that climate means are undergoing significant changes because of global warming [1,2]. Over the past half-century, there have been great changes in precipitation at both global and regional scales [3]. Climate change has a relatively large impact on various aspects of precipitation including total precipitation, precipitation intensity, and the number of rainy days [4]. Precipitation is one of the most important components of the hydrological cycle. Human activities influence

water resource management, with effects on natural disasters (e.g., drought and flood), agricultural productivity, economic development, and the ecological environment [1,5,6]. In the recent past, interest in the spatial and temporal characteristics of precipitation has increased, and numerous researchers have studied changes in regional water cycles and corresponding changes in runoff [7–10]. Studies of precipitation patterns, such as the duration of wet periods, can help us understand how climate change impacts water resource availability [11]. Various indices can be used to describe precipitation changes [1,11–15]. For example, Zhang et al. [13] demonstrated that the precipitation regimes were increasingly homogeneous from the northwest to the southeast of China. Tang et al. [2] indicated that the variability in the annual precipitation on the Loess Plateau was mainly caused by changes in the intensity and frequency of precipitation, with only a minor contribution from changes in the length of the rainy season. Huang et al. [14] showed that the values of a precipitation concentration index were mainly high in the northeast, but low in the southwest, of the Hongshui River Basin. Sarricolea et al. [15] indicated that increases in the precipitation concentration index and daily precipitation concentrations from 1976 to 1994 were linked to the warm phase of the Pacific Decadal Oscillation (PDO) in central-southern Chile.

Recently, research has focused on the number of consecutive wet days (CWDs), known as wet periods (WPs) [11,12,16–18]. WPs, defined as CWDs with daily precipitation amounts greater than 1 mm, can be used to investigate precipitation changes [11,16]. Zhang et al. [12] demonstrated that the maximum number of CWDs in China did not seem to be increasing, but the fractional contribution of shorter consecutive WPs was increasing, suggesting that precipitation was intensifying in China. From their study in Europe, Zolina et al. [16] indicated that longer WPs and higher intensities should have a significant impact on the terrestrial hydrological cycle, including subsurface hydrodynamics, surface runoff, and flooding. Wang et al. [17] showed that, because the number of annual maximum CWDs and corresponding precipitation were decreasing, there was a drying trend in northern parts of the Huai River Basin and drought was intensifying. Huang et al. [18] reported that short duration wet or dry spells comprised a large proportion of the episodes in Sichuan, and that the occurrence and fractional contributions of short-duration WPs were increasing.

While water availability and temperature both limit vegetation growth, water availability is more important. Precipitation, therefore, plays an important role in the geographical distribution of vegetation [19–21]. Climate change (i.e., changes in temperature and precipitation) and human activities also affect changes in the normalized differential vegetation index (NDVI). However, the responses to changes in temperature and precipitation vary by region, type of vegetation, time of year, season, and human activities [22–36]. Zhao et al. [33] showed that the decrease in precipitation was the main contributor (42%) to changes in the NDVI in mid-western farmland areas since the 1990s. Anthropogenic factors such as population density, ecological restoration, and urbanization have had a noticeable effect on the NDVI in the Three Gorges Reservoir region. Cui et al. [34] discovered that NDVI and precipitation were negatively correlated and that climatic factors were the main drivers of NDVI variations in the Yangtze River Basin. Qu et al. [36] demonstrated that the response of vegetation to changes in precipitation was relatively slow because water was plentiful, and that land use changes in the form of ecological restoration projects were the major driver of improved vegetation conditions in the Yangtze River Basin.

Numerous researchers have studied the relationships between changes in precipitation and vegetation in northern and northwestern China [6,37–44]. Li et al. [37], Yang et al. [6], and Qin et al. [42] all reported that significant increases in precipitation were closely associated with the West Pacific Subtropical High (WPSH) and the North America Subtropical High (NASH). The relationship between precipitation and the El Niño-Southern Oscillation (ENSO) varied between different periods and on different time scales [6], and seemed to oscillate over periods of 3, 6, and 11 years [42] in northwestern China. Li et al. [37] showed that the trends in the annual and seasonal precipitation over Gansu were not significant, and, using wavelet analyses, demonstrated that the teleconnection between large scale circulation patterns and summer precipitation varied from region to region, and at different time

scales and over different time periods. Wang et al. [39] also detected that climate indices had changed significantly, with linear trends and abrupt changes noted for all climate indices, with perhaps serious impacts on the water resources and ecological environment in arid regions of China. Wen et al. [40] found that the frequency and intensity of precipitation were increasing in Gansu, and that the WPSH had obvious effects on many precipitation indices measured in two subregions of Gansu. The Indian Ocean Dipole and multivariate ENSO indices may also be important drivers of change in the southeast of China. Gao et al. [41] demonstrated that precipitation decreased by 0.65 mm year$^{-1}$, and the average precipitation intensity increased by 0.2 mm d$^{-1}$ year$^{-1}$, from 1960 to 2012. Wang et al. [44] established that topography and local circulation both influenced the correlation between precipitation and altitude, and that, because of the effects of local precipitation on alpine mountains, the relationship was complex on short timescales, but was statistically significant at monthly, dry/wet season, and annual timescales. Yuan et al. [29,30] found that the growing season NDVI was mainly controlled by temperature in most forest-covered areas of southern China and by precipitation in most grasslands of northern China. Increases in the NDVI were mainly caused by increases in precipitation, although grasslands in northern China responded in different ways to changes in precipitation intensities. Zhao et al. [33] demonstrated that, on the Loess Plateau, the precipitation extremes were significantly and positively correlated with the NDVI during spring and autumn, and negatively correlated with precipitation extremes during winter.

To the best of our knowledge, the relationships between precipitation indices, large-scale circulation patterns, and WPs have not yet been established for the Heihe River Basin. Therefore, the main objectives of this study were (1) to examine temporal and spatial variations in the precipitation indices; (2) to analyze the relationships between monthly precipitation and large-scale circulations; and (3) to identify the relationship between precipitation and the NDVI.

## 2. Study Area

The Heihe River Basin (HRB) originates in the Qilian Mountains, and discharges into Juyanhai Lake. It is located along the land and maritime Silk Roads ("One Belt, One Road") between 97.1° E–102.0° E and 37.7° N–42.7° N, with a total area of approximately $1.43 \times 10^5$ km$^2$ [45–47]. The second largest inland river located in northwest China, the Heihe River is divided into three reaches (upper (UHRB), middle (MHRB), and lower) by two hydrological stations (Yingluoxia and Zhengyixia). In this study, the UHRB and MHRB were selected as the study area. The average annual precipitation exceeds 450 mm and generally increases by 15.5–16.4 mm for every 100-m increase in elevation in the UHRB, and this area is the main source of water flow for the entire river basin. Approximately 90% of water in the MHRB and downstream is replenished from Qilian Mountain [48,49]. Nearly 70% of the total precipitation occurs between June and September, with only 3.5% falling between December and February [49,50]. Annual precipitation ranges from 69 to 216 mm in the MHRB, and this area consumes large amounts of water via a relatively complete agricultural irrigation system consisting of more than 893 main canals and branch canals [47]. Over 90% of the population, grain production and major industries are concentrated in the MHRB. Approximately 84% of the total available water is consumed for irrigation, with demand consistently increasing [51,52]. Land use/cover (LUCC) has also significantly changed in the basin. Built land, cultivated land and forests have increased, while water areas and grasslands have become increasingly degraded [52,53]. Irrigation, dam construction and other types of water diversion engineering are also intensive in the MHRB [52]. The study area location and meteorological station information is shown in Figure 1 and Table 1.

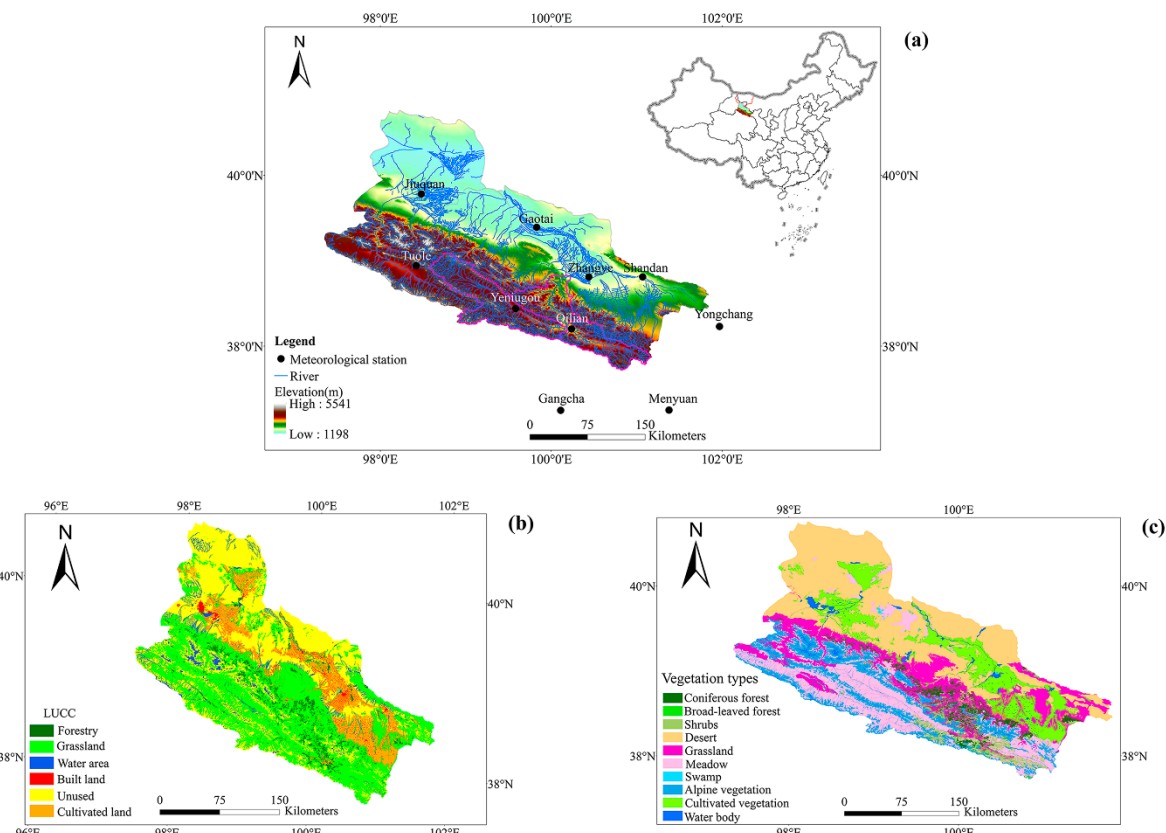

**Figure 1.** Location and meteorological stations (**a**), land use/cover (LUCC) for 2011 (**b**), and vegetation for 2011 (**c**) in the upper (UHRB) and middle (MHRB) reaches of the Heihe River Basin.

**Table 1.** Details on the meteorological stations in the study area.

| Station | Longitude (E) | Latitude (N) | Elevation (m) | ATP | ATD | API | AMRD |
|---------|---------------|--------------|---------------|------|------|------|------|
| Qilian | 100.24° | 38.19° | 2787.4 | 368.13 | 70.82 | 10.11 | 0.10 |
| Yeniugou | 99.58° | 38.41° | 3286.0 | 270.98 | 75.73 | 10.00 | 0.10 |
| Tuole | 98.01° | 39.03° | 3283.0 | 374.78 | 52.96 | 9.23 | 0.08 |
| Gangcha | 100.11° | 37.25° | 3309.0 | 354.59 | 67.46 | 9.40 | 0.10 |
| Menyuan | 101.38° | 37.25° | 2867.0 | 475.45 | 85.73 | 10.40 | 0.13 |
| Gaotai | 99.79° | 39.36° | 1332.2 | 97.30 | 23.32 | 5.70 | 0.05 |
| Shandan | 101.08° | 38.77° | 1764.6 | 179.87 | 36.71 | 7.25 | 0.07 |
| Yongchang | 101.58° | 38.18° | 1976.1 | 184.98 | 39.43 | 6.98 | 0.07 |
| Jiuquan | 98.49° | 39.70° | 1477.2 | 78.61 | 19.86 | 5.26 | 0.04 |
| Zhangye | 100.46° | 38.91° | 1482.7 | 114.08 | 26.77 | 5.86 | 0.05 |

NB: the definitions of abbreviations are shown in Table 3.

## 3. Data and Methods

### 3.1. Data

Daily precipitation data were collected from 10 national standard meteorological stations (five stations in the UHRB, including two nearby stations, and five stations in the MHRB, including one nearby station) spanning from 1961 to 2016 (Figure 1). The data were derived from the China Meteorological Data Sharing Service System of the National Meteorological Information Center V3.0 (http://www.nmic.gov.cn/). Rigorous quality control was conducted by the China National Meteorological Information Center before the data were released. The software used to detect and adjust shifts in the time series of daily precipitation were RHtestsV3 and Rhtests-dlyPrcp (http://etccdi.pacifcclimate.org/software.shtml), respectively. Shuttle Radar Topography Mission (SRTM)

elevation data with a resolution of 90 m were derived from the Geospatial Data Cloud for China (http://www.gscloud.cn/). The monthly multivariate ENSO index (http://www.esrl.noaa.gov/psd/enso/mei/index.html) was used to characterize ENSO events from 1961–2016. The monthly Atlantic Oscillation (AO) from the National Oceanic and Atmospheric Administration (NOAA) of the National Climatic Data Center (http://www.ncdc.noaa.gov/teleconnections/ao.php), the monthly PDO indices from the Tokyo Climate Center (ds.data.jma.go.jp/tcc/tcc/products/elnino/decadal/annpdo.txt), the monthly North Arctic Oscillation (NAO) indices from the NOAA of the National Geophysical Data Center website (http://www.ngdc.noaa.gov), the monthly Atlantic Multidecadal Oscillation (AMO) indices from the NOAA of the Earth System Research Laboratory (www.esrl.noaa.gov/psd/data/correlation/amon.us.data), and the monthly Pacific North American teleconnection pattern (PNA) indices from the NOAA of the Climate Prediction Center (http://www.cpc.ncep.noaa.gov/data/teledoc/pna.shtml) were used to represent large-scale climate anomalies correlated with precipitation from 1961 to 2016. Precipitation indices, concepts, and related units are shown in Table 2. The annual and summer (June-July-August) NDVI spatial distribution dataset was remote sensing data from SPOT/VEGETATION NDVI and MODIS based on a continuous time series and was obtained from the resource and environmental science data center of the Chinese Academy of Sciences (http://www.resdc.cn). A 1:100,000 vegetation map of the Heihe River Basin (version 2.0) in the UHRB and MHRB, and land use/cover data for 1975, 1985, 1995, 2000, 2010, and 2011 in the MHRB were obtained from a scientific data center for cold and arid regions (http://westdc.westgis.ac.cn/).

### 3.2. Methods

#### 3.2.1. Theil-Sen Median Trend Analysis and Mann–Kendall Test Statistic

The Theil-Sen median trend [54,55] analysis method was used to estimate the trend slopes of the precipitation time series. The Mann–Kendall (M–K) nonparametric statistical test method recommended by the World Meteorological Organization (WMO), which was proposed by Mann [56] and Kendall [57], has been widely used in meteorology and hydrological mutation testing. Here, the M–K test was applied to analyze the temporal characteristics of precipitation. For a time series $x$ with a sample size of $n$, assuming that the original time series is random and independent, the test statistic $Z_{mk}$ is calculated as:

$$Z_{mk} = \begin{cases} \frac{S-1}{\sqrt{Var(S)}}, & S > 0 \\ 0, & S = 0 \\ \frac{S+1}{\sqrt{Var(S)}}, & S < 0 \end{cases} \tag{1}$$

where,

$$S = \sum_{i=1}^{n-1} \sum_{k=i+1}^{n} \text{sgn}(x_k - x_i) \tag{2}$$

$$Var(S) = \frac{\left[ n(n-1)(2n+5) - \sum_{i=1}^{m} t_i(t_i-1)(2t_i+5) \right]}{18} \tag{3}$$

where: $Var(S)$ is the variance of the statistic $S$; $x_k$ and $x_i$ are the sequential data values; $m$ is the number of tied groups; $t_i$ denotes the number of data points in the $i$ th group; $n$ is the length of the data set; and sgn $(x_k - x_i)$ is the sign function, determined as

$$\text{sgn}(x_k - x_i) = \begin{cases} +1, & (x_k - x_i) > 0 \\ 0, & (x_k - x_i) = 0 \\ -1, & (x_k - x_i) < 0 \end{cases} \tag{4}$$

For the $Z_{mk}$ value, $Z_{mk} > 0$ indicates that the time series has a rising (increasing) trend while a time series with $Z_{mk} < 0$ shows a falling (decreasing) trend. Absolute values of $Z_{mk} \geq 1.65$, 1.96, 2.58, were adopted to indicate significance levels of $\alpha = 0.1$, 0.05, and 0.01, respectively.

When the M–K test is used to test the sequence mutation, the test statistic differs from the $Z_{mk}$ given above, through the construction of a rank sequence, as follows:

$$S_k = \sum_{i=1}^{i} \sum_{j}^{i-1} \alpha_{ij} \qquad (k = 2, 3, 4, \dots, n) \tag{5}$$

$$\alpha_{ij} = \begin{cases} 1 & x_i > x_j \\ 0 & x_i < x_j \end{cases} \qquad 1 \leq j \leq i \tag{6}$$

$$UF_k = \frac{[S_k - E(S_k)]}{\sqrt{Var(S_k)}} \qquad (k = 1, 2, \dots, n) \tag{7}$$

$$E(S_k) = k(k+1)/4 \tag{8}$$

$$Var(S_k) = k(k-1)(2k+5)/72 \tag{9}$$

where $UB_k$ is a standard normal distribution, and a significant level $\alpha$ is given. If there is a significant trend change, the time series $x$ is arranged in reverse order, and then calculated according to the formula:

$$\begin{cases} UB_k = -UF_k \\ k = n+1-k \end{cases} \qquad (k = 1, 2, \dots, n) \tag{10}$$

where $UF_k$ is a positive sequence and $UB_k$ is a reverse sequence. If $UF_k$ exceeds 0, the sequence shows a rising trend, and a value of <0 indicates a falling trend. The rising or falling trend is significant when these parameters exceed the critical thresholds. If the $UF_k$ and $UB_k$ curves intersect, and the intersection is between the critical thresholds, the corresponding moment of intersection is defined as the moment when the mutation begins.

### 3.2.2. Wavelet Coherence

Wavelet coherence (WTC) was used to analyze the coherence of the cross-wavelet spectrum in the time–frequency space and thus, it can be used to determine the intensity of the covariance of the two time series $x_t$ and $y_t$. It is an appropriate and powerful tool to characterize intermittent cross-correlations between two time series at multiple time scales [58–62]. The WTC of two time series $X$ and $Y$ can be calculated by:

$$R_t^2(s) = \frac{\left| S(s^{-1} W_t^{XY}(s)) \right|^2}{S(s^{-1} |W_t^X(s)|^2) S(s^{-1} |W_t^Y(s)|^2)} \tag{11}$$

where $R_t^2(s)$ is a squared cross-WTC, $t$ is a time index, $s$ denotes the time scale, $W_t^{XY}(s)$ represents the amount of joint power between $X$ and $Y$, and $S$ represents a smoothing operator that can be expressed as:

$$S(W) = S_{scale}(S_{time} W_t(s)) \tag{12}$$

where $S_{scale}$ is the smoothing along the wavelet scale and $S_{time}$ is the smoothing in time. The value of squared WTC is in the range from 0 to 1. If its value is close to 1, strong interdependence will be identified and vice versa. For the Morlet wavelet, a suitable smoothing operator is given by Torrence and Compo [62] as follows:

$$S_{time}(W)|_s = \left( W_t(s) c_1^{\frac{-t^2}{2s^2}} \right) \tag{13}$$

$$S_{time}(W)_t|_t = (W_t(s)c_2 \prod (0.6s)|_t \tag{14}$$

where $c_1$ and $c_2$ are normalization constants, $\prod$ denotes the rectangle function; and 0.6 is an empirically determined scale decorrelation length for the Morlet wavelet [62]. The statistical significance level of the WTC is estimated using Monte Carlo methods [62,63].

### 3.2.3. Other Methods

The Pearson, Kendall, and Spearman tests were used to determine the lagged correlations between precipitation and climate indices. Pearson's *r* was used to explore the relationships between the annual NDVI and precipitation. The *t*-test was used to assess the statistical significance (*p*) of the slope and *r* [30]. A kriging method was used to interpolate the annual precipitation for the period from 1998 to 2015.

## 4. Results and Discussion

### 4.1. Occurrence and Fractional Contribution of WPs

As shown in Figure 2, the occurrence and fractional contribution (which is consecutive 1, 2, . . . , *n* wet days (precipitation) divided by the total number of wet days (total precipitation)) of WPs decreased exponentially as the duration of the wet periods increased. Occurrence accounted for 56.3% and 75.1% of precipitation in the UHRB and MHRB, respectively, and 1-day WPs dominated. WPs with durations of more than 9 days accounted for only about 0.2% of the events in the UHRB and WPs with durations of more than 5 days accounted for approximately 0.3% in the MHRB. The fractional contribution of WPs to the total precipitation was similar to the occurrence of WPs, and 1-day WPs accounted for 27.8% and 52.6% of precipitation in the UHRB and MHRB, respectively. The fractional contribution of 1–4-day WPs was 82.7% in the UHRB, and of 1–2-day WPs was 82.1% in the MHRB. Therefore, 1–4-day WPs in the UHRB and 1–2-day WPs in the MHRB accounted for the largest fractional contributions to the annual total precipitation. These findings were consistent with those of Zolina et al. [16] and Zhang et al. [11,12], who found that 1-day WPs dominated occurrence, but were not consistent with Zolina et al. [16] and Zhang et al. [11,12] who reported that the fractional contribution of 2-day-WPs was the largest.

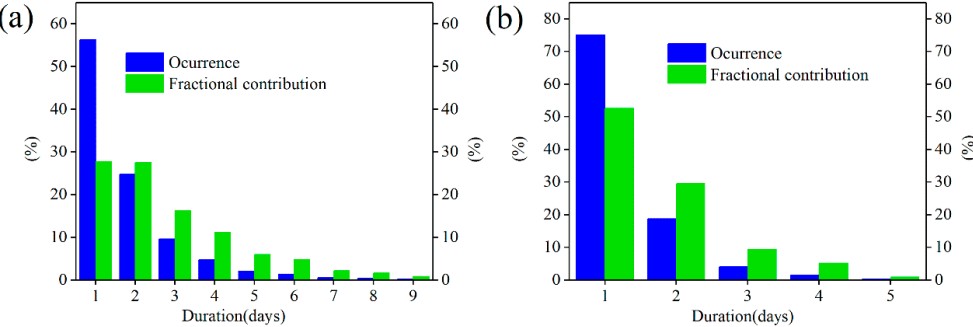

**Figure 2.** Occurrences and fraction contributions of wet period (WP) durations in the upper (UHRB) (**a**) and middle (MHRB) (**b**) reaches of the Heihe River Basin.

Zhang et al. [12] pointed out that a higher occurrence and fractional contribution of 1-day-WPs in northwest China led to a higher frequency of flash floods and droughts in this region. In this study, WPs occurred more frequently in the UHRB than in the MHRB, but the precipitation concentration was higher in the MHRB than in the UHRB. The risk of flash floods was higher in the UHRB than in the MHRB because of the high occurrence/fractional contribution of 1-day WPs (56.3%/27.8%). The occurrence/fractional contribution of 1-day WPs accounted for 75.1%/52.6% in the MHRB, where they

may have increased urban waterlogging (over 90% of the population and major cities are concentrated in the MHRB) and also may have enhanced the drought risk.

*4.2. Temporal Changes in the Occurrence and Fractional Contribution of WPs*

Figure 3 shows the temporal patterns in the normalized occurrence and fractional contribution. The normalized occurrence and fractional contribution were derived as $Z = (X - \overline{X})/\mathrm{Std}(x)$, where $Z$ is the normalized series, $X$ represents the occurrence and fractional precipitation series, $X$ is the average value of $X$, and $\mathrm{Std}(x)$ denotes the standard deviation of $X$. All the normalized scores were smoothed by a 5-year running average obtained from the meteorological stations over the UHRB and MHRB. As shown in Figure 3a,b, before the 1990s (especially in the 1960s–1970s), 5–8-day WPs were the most frequent, and after the 2000s, 2–3-day WPs were the most frequent in the UHRB. In the MHRB, 2–3-day WPs occurred most frequently between 1975 and 1980, 4–5-day WPs dominated from 1980 to 1985, and 1-day WPs dominated after the 2000s. These results were different from the findings of Zolina et al. [16] in their study in Europe, but, although the duration days differed on the interdecadal scale, they were almost consistent with the findings of Zhang et al. [13]. The results imply that different regions respond differently to changes in precipitation driven by global climate change.

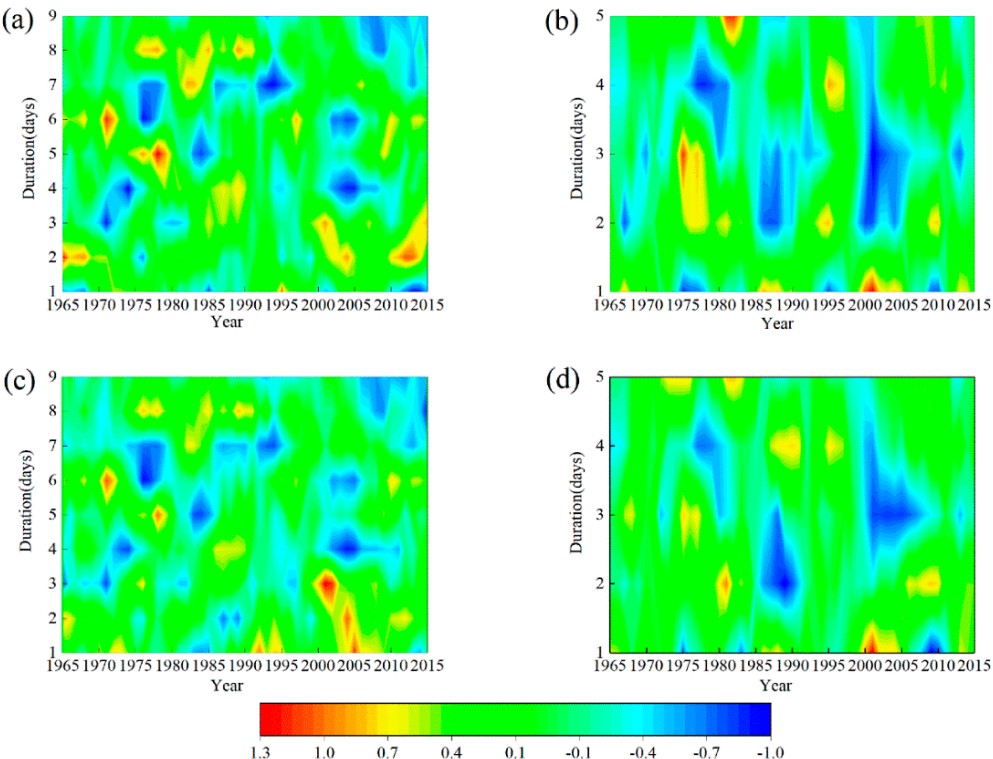

**Figure 3.** Temporal evolution of normalized occurrences (**a**) denotes the upper (UHRB) and (**b**) the middle (MHRB) reaches of the Heihe River Basin and fractional contribution (**c**) denotes UHRB, (**d**) denotes MHRB anomalies in different wet periods (WPs) in the UHRB and MHRB.

The temporal pattern in the fractional contributions of WPs was similar to that of the occurrence of the WPs. Figure 3c shows that 5–6-day WPs during the 1960s and 7–8-day WPs during the 1970s and 1980s made the greatest fractional contributions to the annual total precipitation. Since 2000, 1–2-day WPs have made the largest fractional contribution to the annual total precipitation in the UHRB. The fractional contributions of WPs were consistent with occurrence in the MHRB. Hence, the occurrences and fractional contributions of WPs of longer durations decreased, whereas shorter durations increased, especially after the 1990s, which means that shorter WPs occurred more frequently in the UHRB and MHRB. Zhang et al. [11] pointed out that there were episodes when precipitation and hydrological

cycling were intensifying in the Pearl River Basin after the early 1980s; similarly, in the UHRB and MHRB, the precipitation episodes and hydrological cycle also intensified after the early 2000s.

*4.3. Spatial Distribution of Normalized Occurrences and Fractional Contributions for WPs*

Figures 4 and 5 show the M–K trend for the *Z* value of the spatial distribution of normalized occurrences and fractional contributions of WPs with different durations, i.e., 1–2 days; 3–5 days in 10 stations before and after an abrupt change point, total period 1961–2016. The significance of trends is evaluated at the 0.01 level. Figure 4a shows that, before abrupt change points that took place at various times, the occurrences of 1-day-WPs and 4-day-WPs for 8 and 6 stations increased, and accounted for 80% and 60% of total precipitation stations, respectively; while 2–3 day-WPs and 5-day WPs for 4 and 7 stations decreased, and accounted for 40% and 70% of the total precipitation stations, respectively. As shown in Figure 4b, after an abrupt change point, the occurrences of 1-, 3-, and 4-day-WPs for 6 stations increased, and accounted for 60% of the total precipitation stations, respectively, while durations of 2- and 5-day-WPs for 7 and 5 stations decreased, accounting for 70% and 50% of the total precipitation stations, respectively. For the whole period, 1-day-WPs and 3–4-day WPs for 5 (50%) and 6 (60%) stations increased, respectively, while 2-day- and 5-day-WPs for 6 (60%) and 5 (50%) stations decreased, respectively (Figure 4c).

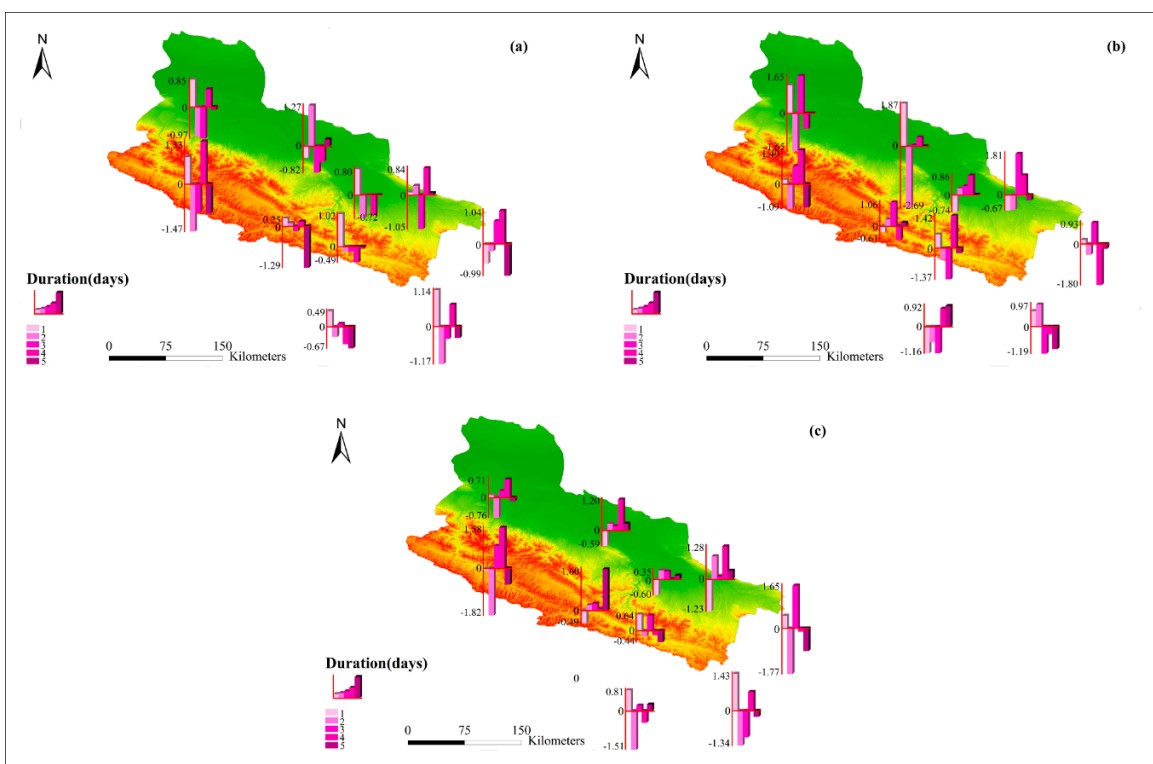

**Figure 4.** Spatial distribution of normalized occurrences of Mann–Kendall *Z* values for different wet periods (WPs) before (**a**) and after (**b**) abrupt changes and over the total period (**c**) over the upper (UHRB) and the middle (MHRB) reaches of the Heihe River Basin.

The spatial distributions of the fractional contributions and occurrences of WPs were similar (Figure 5). Before an abrupt change point, the fractional contributions of 1-, 3-, and 4-day WPs increased at 6 of the 10 stations, respectively, across the UHRB and MHRB (Figure 5a). The fractional contributions of 4- and 5-day WPs decreased for 5 and 8 of the 10 stations, respectively, across the UHRB and MHRB. The fractional contributions of 1–3-day WPs decreased at 3 stations, but increased at 2 stations, accounting for 30% and 20% of the total precipitation stations, respectively. At the same time, 4-day-WPs increased at 6 stations while 5-day-WPs decreased at 6 stations after an abrupt change

point (Figure 5b). Over the whole time period, the fractional contributions of 1–2-day and 4–5-day WPs decreased at 4 and 2 stations, respectively, and the fractional contributions of 3–4-day WPs increased at 5 and 2 stations, respectively (Figure 5c).

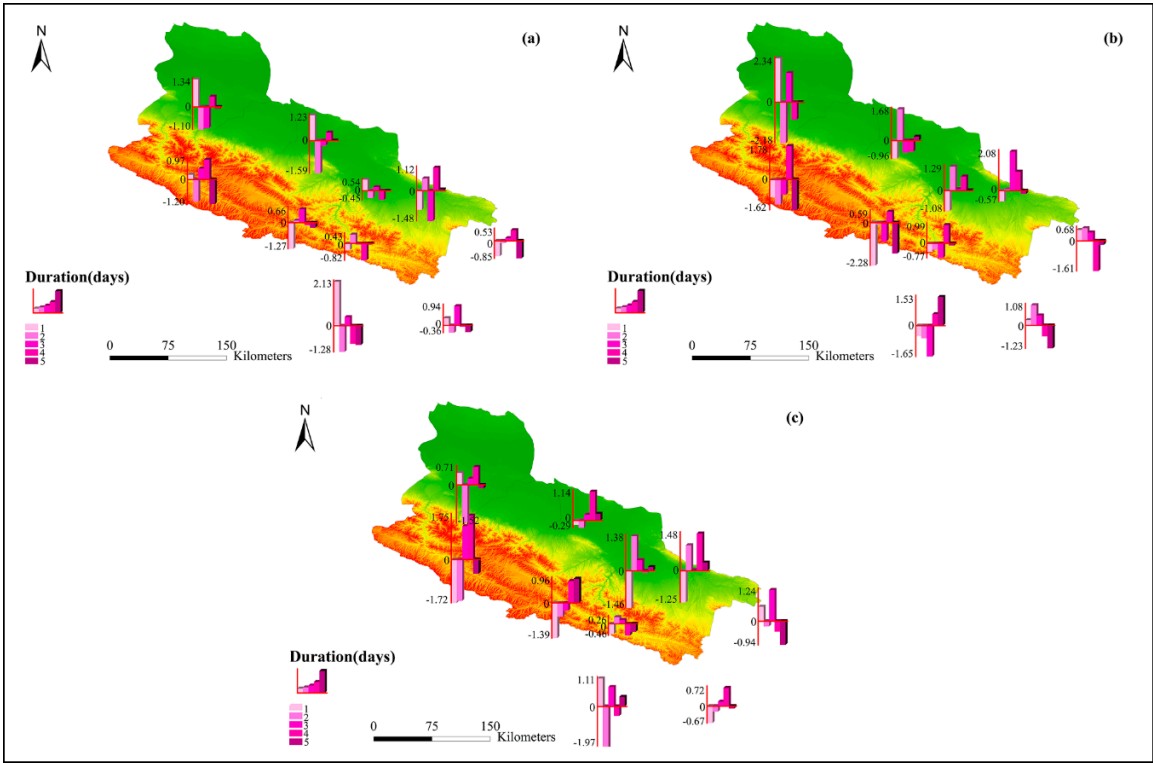

**Figure 5.** Spatial distribution of normalized fraction of Mann–Kendall *Z* values for different wet periods (WPs) before (**a**) and after (**b**) abrupt changes and over the total period (**c**) over the upper (UHRB) and the middle (MHRB) reaches of the Heihe River Basin.

The above analysis shows that the spatial distribution of occurrences and fractional contributions were almost consistent with the spatial distribution patterns, and that the significant increases and decreases for the fractional contributions and occurrences were sporadic for some stations of WPs (i.e., 1-day, 2-days) in the UHRB and MHRB.

*4.4. Trends in Precipitation Indices*

Figure 6 and Table 2 show the evolution and statistical changes in precipitation indices in the UHRB and MHRB. In the UHRB and MHRB, the annual average ATP was 368.8 and 130.9 mm from 1961 to 2016, and changed at rates of 1.04 mm year$^{-1}$ (Z = 3.09) and 0.43 mm year$^{-1}$ (Z = 2.16), with abrupt change points in 2004 and 1981 in the UHRB and MHRB, respectively. The significant increases in the ATP are consistent with those reported for northwestern Gansu Province, the Tianshan Mountains, and northwestern China [37–40,42], but differ from reported decreases on the Chinese Mongolian Plateau [41] (Figure 6a,e). The long-term annual average ATD in the UHRB was 70.5 days and changed at a rate of 0.05 day year$^{-1}$ (Z = 0.93), and in the MHRB it was 29.2 days and changed at a rate of 0.086 day year$^{-1}$ (Z = 2.24), with an abrupt change point in 2000 (Figure 6b,f). The long-term annual average API was 9.8 mm/day, with a rate of change of 0.023 mm/day year$^{-1}$ (Z = 3.15) and an abrupt change point in 1976, in the UHRB, and it was 6.2 mm/day, with a rate of change of 0.007 mm/day year$^{-1}$ (Z = 0.87) (Figure 6c,g) in the MHRB. The long-term annual average AMRD was 2.3 days in the UHRB and changed at a rate of 0.002 days year$^{-1}$ (Z = 0.94), and was 1.0 days and changed at a rate of 0.003 days year$^{-1}$ (Z = 2.23) in the MHRB, with an abrupt change point in 1992 (Figure 6d,h). As shown in Table 2, the ATP, ATD, API, and AMRD (see Table 3 for definitions) increased at all stations except

the ATD in Menyuan and Gaotai, and the AMRD in Gangcha. The ATP increases were significant at 5 stations. The ATD increases were statistically significant at 3 stations (Table 2). The API and AMRD had significant increases at 2 and 3 stations, respectively.

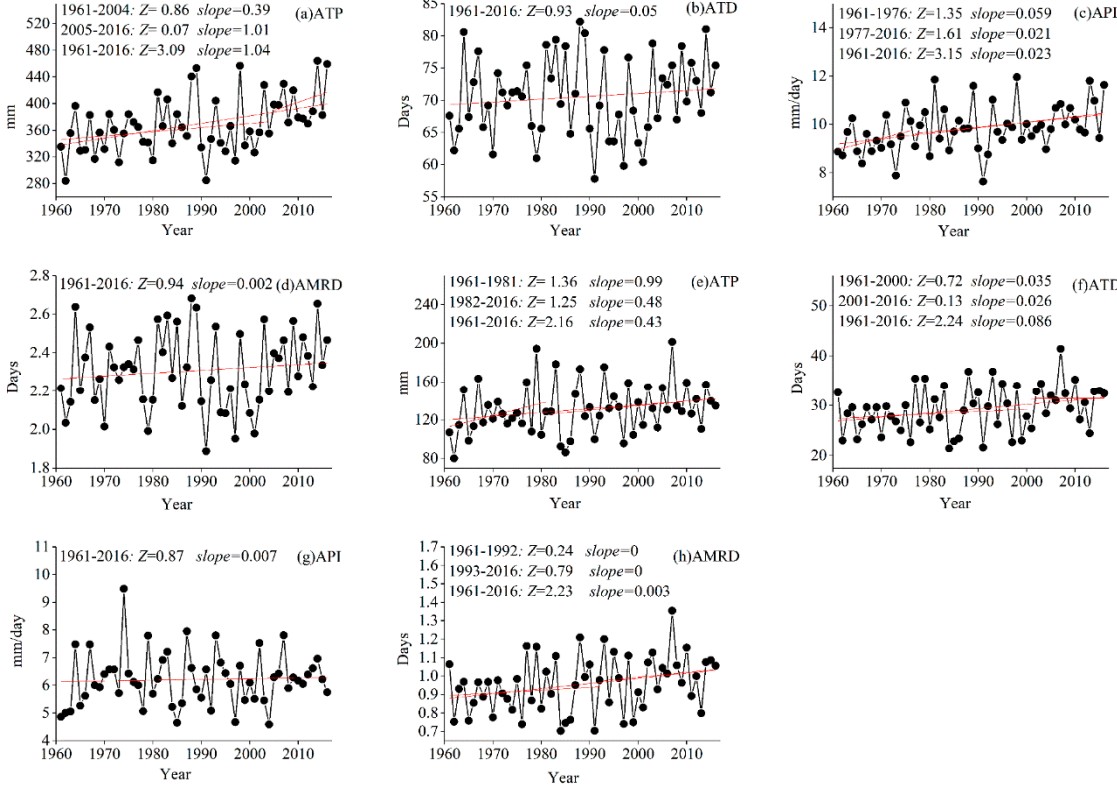

**Figure 6.** Trend of annual average precipitation indices before and after abrupt change points and over the total period from 1961 to 2016 in the upper (UHRB) (**a–d**) and middle (MHRB) reaches of the Heihe River Basin (**e–h**). (Solid line denotes the linear trend, shading denotes the 95% confidence intervals of the estimated trends).

**Table 2.** The $Z$ value of the precipitation indices, 1961–2016.

| Station | ATP | ATD | API | AMRD |
|---|---|---|---|---|
| Qilian | 1.96 [a] | 0.35 | 1.48 | 0.91 |
| Yeniugou | 3.56 [b] | 1.82 | 2.72 [b] | 1.61 |
| Tuole | 3.27 [b] | 2.11 [a] | 2.44 [a] | 1.99 [a] |
| Gangcha | 2.18 [a] | 0.66 | 1.51 | −0.76 |
| Menyuan | 0.88 | −1.31 | 1.34 | 0.93 |
| Gaotai | 1.28 | 2.7 [b] | −0.05 | 2.3 [a] |
| Shandan | 1.36 | 1.53 | 1.86 | 0.42 |
| Yongchang | 2.27 [a] | 2.62 [b] | 0.29 | 2.45 [a] |
| Jiuquan | 1.18 | 1.78 | 0.25 | 1.77 |
| Zhangye | 0.49 | 0.44 | 0.54 | 0.37 |

Significance test by MK (Mann–Kendall), [a] Significance at 0.05 confidence level ($Z \geq 1.96$), [b] Significance at 0.01 confidence level ($Z \geq 2.58$). See Table 3 for definitions.

Increases in the ATP and ATD caused the API in the UHRB and MHRB to increase and, at the same time, the AMRD also increased. Gao et al. [41] found that the total precipitation and the number of rainy days on the Chinese Mongolian Plateau were decreasing but that the average precipitation intensity was increasing.

*4.5. Changes in the Attributes of the Precipitation Regimes*

The changes in the precipitation indices, as listed in Table 3, are presented in Figure 7. The SP/A, SU/A, AU/A, and CV precipitation indices decreased after abrupt changes in the UHRB and MHRB. However, the W/A, SP/SU, SU/AU, and AU/W increased, indicating increases in the monthly precipitation in winter, and possibly decreases in the monthly precipitation in summer and autumn, which suggests that the dry seasons were possibly becoming wetter, and the wet seasons were becoming drier. These results are consistent with those of Zhang et al. [11,13], who found that, in most parts of China, the autumn precipitation was decreasing, the winter precipitation was increasing, and the total precipitation of the maximum CWDs was increasing annually and in winter, implying that there was a possibly trend towards wetting in northwest China and in winter. The changes in precipitation in autumn are complicated, but Figure 7 shows that the precipitation changes were markedly different in the UHRB and MHRB. Also, the slight changes in the MAX, MIN, and CI indicated that the monthly average maximum precipitation and monthly average minimum precipitation regimes did not shift.

**Table 3.** Definitions of precipitation indices (Zhang, et al. [11,13]).

| Precipitation Indices | Definitions | Units |
|---|---|---|
| ATP | Annual total precipitation amount when precipitation ≥1 mm | mm |
| ATD | Annual total rainy days | day |
| API | Annual precipitation intensity | mm/day |
| AMRD | Annual mean rainy days | day |
| W/A | The ratio between the average of the total monthly precipitation during December and February of the next year and the total annual precipitation | % |
| SP/A | The ratio between the average of the total monthly precipitation during March and May of the next year and the total annual wet precipitation | % |
| SU/A | The ratio between the average of the total monthly precipitation during June and August of the next year and the total annual wet precipitation | % |
| AU/A | The ratio between the average of the total monthly precipitation during September and November of the next year and the total annual wet precipitation | % |
| W/SP | Ratio between winter precipitation and spring precipitation | % |
| SP/SU | Ratio between spring precipitation and summer precipitation | % |
| SU/AU | Ratio between summer precipitation and autumn precipitation | % |
| AU/W | Ratio between autumn precipitation and winter precipitation | % |
| MAX | Ratio between the maximum monthly average precipitation and total annual precipitation | % |
| MIN | Ratio between the minimum monthly average precipitation and total annual precipitation | % |
| CI | Ratio between the minimum and the maximum monthly average precipitation | % |
| CV | Ratio between the standard deviation and the average monthly precipitation | % |

*4.6. Wavelet Coherence between the Monthly Precipitation and Large-Scale Climate Indices*

As shown in Figure 8a, the effect of ENSO was weak, relatively dispersed, and did not last for a long period on the scale of 1–12 months. Conversely, precipitation lagged behind the ENSO 7.8–13.1 months' signal on the scale of 31.2–52.5 months from 1969 to 1975 and had a strong significant

and negative influence on precipitation at the 19.7–26.2 month scale from 1982 to 1987 (excluding restricted areas, hereafter it was the same). It also had a strong significantly negative advanced correlation on the scale of 18.6–32.8 months from 1993 to 2000 (precipitation advanced ENSO from 4.7 to 8.2 months), which is consistent with an ENSO periodicity of 2 to 7 years [64]. As shown in Figure 8b, there was a lag of between 10.4 and 15.6 months on the scale of 41.6–62.4 months from 1966 to 1974 between precipitation and PDO, while on the 49.5–70-month scale, precipitation preceded PDO by 12.4–17.5 months from 1982 to 1996. There was a strong significant positive/negative, advanced correlation between PDO and precipitation on the scale of 7.8–15.6 months from 1988 to 2002, while on the 8.2–14.7 month scale, there was a strong significant negative correlation from 2010 to 2015. As shown in Figure 8c, the PNA was relatively dispersed, with significant advanced/lagged, and positively/negatively correlations with precipitation on the 4–16 months scale over the whole time period, while on the scale of 41.6–55.6 months, the signal had a significant lagged correlation (precipitation lagged PNA by between 10.4–13.9 months) from 1966 to 1973. Figure 8d shows that the NAO and precipitation were significantly and positively correlated from 1962 to 1972 and from 1965 to 1972 at scales of 8.8–16.5 and 19.8–35 months, respectively. There was also a significant lagged (precipitation lagged NAO by 6.5–8 months) correlation between precipitation and NAO on the scale of 29.4–41.6 months from 1996–2002. Figure 8e shows that AO and precipitation were significantly and positively correlated at scales of 9.3–16.5 months and 44.1–78.6 months from 1962 to 1967 and from 1978 to 1995, respectively. Meanwhile, on the scale of 49.5–58.9 months, there was a significant advanced correlation between AO and precipitation from 1968–1976, in which AO preceded precipitation by 12.4–14.7 months. As shown in Figure 8f, AMO and precipitation were significantly and negatively correlated on a scale of 39.3–52.5 months from 1970 to 1975, and were significantly and positively correlated on the scale of 7.8–15.6, 9.3–14.7, and 7.4–16.5 months from 1985 to 1990, 1997 to 2000, and 2001 to 2015, respectively.

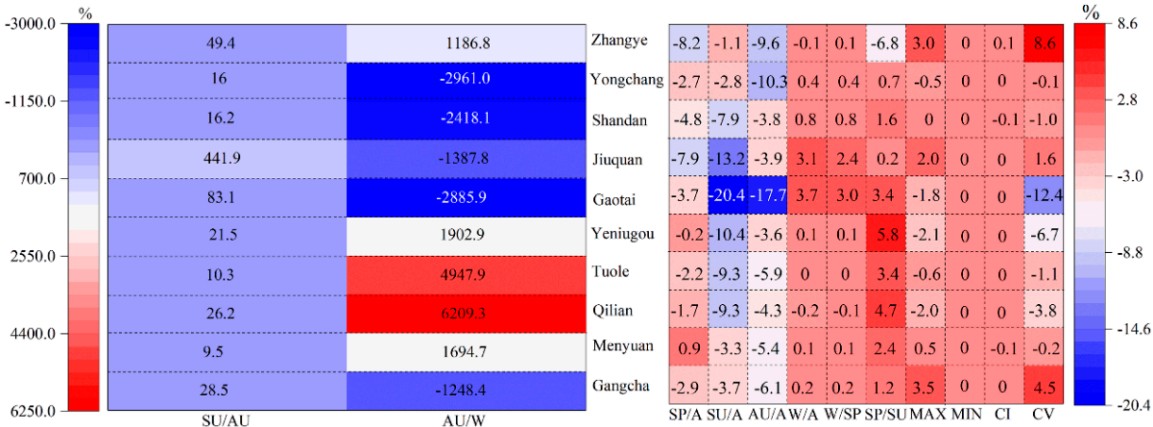

**Figure 7.** Temporal changes of 12 precipitation variable metrics after abrupt shifts.

There was a significant advanced/lagged positive/negative correlation between ENSO and precipitation on the scale of 1–12 months (Figure 9a), but it was relatively dispersed, and it did not have a long impact period. However, there were significant advanced correlations between precipitation and ENSO at three time scales (from 49.5–70, 19.7–29.4, and 19.2–27.8 months) over three time periods (from 1977 to 1989, 1995 to 1997, and from 2007 to 2009). Precipitation advanced PDO by 2.4–3.5 months on the scale of 9.5–14 months from 1994 to 1998 (Figure 9b), and precipitation and PDO were significantly and positively correlated at scales of 8.8–12.4 months and 8.3–14.7 months from 1971–1975 and 1988–1991, and significantly and negatively correlated at the 7.4–15.6 month-scale from 2010–2015. There was a relatively dispersed significant advanced/lagged, positive/negative correlation between PNA and precipitation on the 4–16 month-scale over the whole time period, and significant positive correlations on scales of 37.4–45.8 months and 24.8–44.1 months from 1979 to 1985 and from 2000 to

2013, respectively (Figure 9c). There was also a significant positive advanced correlation between PNA and precipitation on the 18.6–25.2 month-scale from 2005 to 2013. The NAO and precipitation were significantly and positively correlated at scales of 8.8–18.6, 19.7–37.1, and 8.8–14.7 months from 1962 to 1972, 1964 to 1972, and from 1974 to 1978, respectively (Figure 9d). There was also a significant lagged correlation between precipitation and NAO on the 41.6–52.5 month-scale from 1997 to 2003, when precipitation lagged NAO by between 9.5 and 11 months. Precipitation and AO were significantly and positively correlated on scales of 11–15.6, 26.2–41.6, and 49.5–58.9 months from 1962 to 1966, 1964 to 1969, and from 2004 to 2010, respectively (Figure 9e). Precipitation and AMO were significantly and positively correlated at scales of 8.8–15.6, 10.4–15.6, and 8.1–14.5 months from 1985 to 1990, 1997 to 2000, and from 2002 to 2015, respectively (Figure 9f).

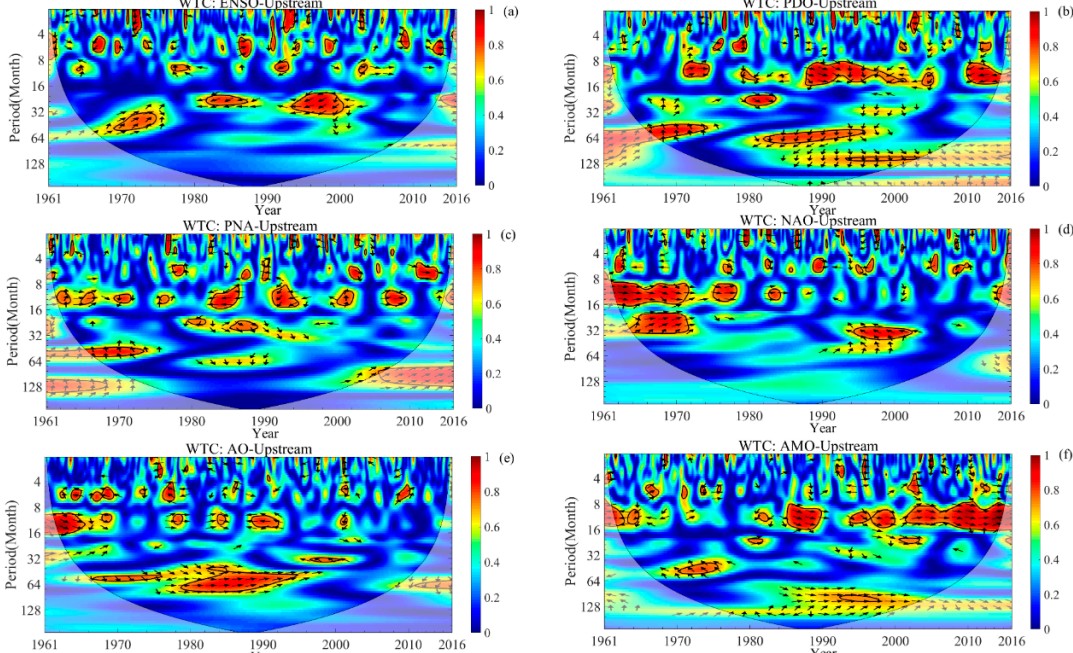

**Figure 8.** The wavelet coherence spectra between monthly observed average precipitation and El Niño-Southern Oscillation (ENSO) (**a**), Pacific Decadal Oscillation (PDO) (**b**), Pacific North American teleconnection pattern (PNA) (**c**), North Arctic Oscillation (NAO) (**d**), Atlantic Oscillation (AO) (**e**), and Atlantic Multidecadal Oscillation (AMO) (**f**) indices covering 1961–2016 in the upper Heihe River Basin (UHRB). The 5% significance level against red noise is exhibited as a thick contour, and the relative phase relationship is denoted by arrows (The arrows pointing to the right for positive correlation and to the left for negative correlation. The arrows pointing vertically downward or upward represent the advanced or lagged correlation with precipitation and the climate indices of the 1/4 cycle).

When the UHRB and MHRB were compared using wavelet coherence analysis plots (Figures 8 and 9), we found that the correlations between the precipitation and climate indices in the UHRB and MHRB were very similar for ENSO, AMO, and NAO, and were relatively short and time period-delayed in the MHRB. The precipitation responses to the time characteristics of the PDO, PNA, and AO indices were not very similar. The responses of the signal and time to the climate indices were more obvious for the UHRB than for the MHRB.

To improve insights into the correlations between precipitation and the climate indices, Pearson, Kendall, and Spearman's correlation coefficients were used to determine the lag time between the climate indices and precipitation in the UHRB and MHRB. Here, according to Pearson, Kendall, and Spearman methods, we took the largest of two of the correlation coefficients, and the significant levels were 0.05/0.01). Precipitation with a lag-time of between 0–12 months was not significantly correlated with ENSO, AO, and PNA in the UHRB (Table 4). However, the Pearson, Kendall, and Spearman's

tests showed that the NAO was significantly and positively correlated with precipitation at 0 month (no lag). We also found that there were significant positive correlations between precipitation and PDO for lags of 3 months. The Pearson, Kendall, and Spearman's correlation coefficients showed significant and positive correlations between AMO and precipitation for lags of 1 month in the UHRB. There were significant positive correlations coefficients between NAO and precipitation at lags of 12 months for Kendall and Spearman. The Pearson, Kendall, and Spearman correlation coefficients also showed significant and positive correlations between the PDO and precipitation at lags of 3 months. The AMO was significantly and positively correlated with precipitation for lags of 1 month for Pearson, Kendall, and Spearman's correlation coefficients in the MHRB. However, when correlations are low, low correlation coefficients with significant *p*-values may occur due to the large sample size used in this study, which resulted in high degrees of freedom [65]. Furthermore, De Oliveirajunior et al. [66] revealed that all tests, regardless of the Standardized Precipitation Index scale, showed low Pearson, Kendall, and Spearman correlation coefficients, while significant for ENSO and PDO in the north and northwest regions of the State of Rio de Janeiro, Brazil. It was concluded that the ENSO and PDO signals are unclear and, therefore, do not influence precipitation independently. The analysis demonstrated low significance Pearson, Kendall, and Spearman lag correlation coefficients for precipitation and the climate indices. This suggests that there may be a weak lag linkage between precipitation and climate indices (or precipitation with climate indices signals are unclear), in addition factors such as high degrees of freedom and the fact that climate indices do not affect precipitation independently. Furthermore, the correlation coefficients of the three methods are different, which may be related to the different calculation methods of the three approaches.

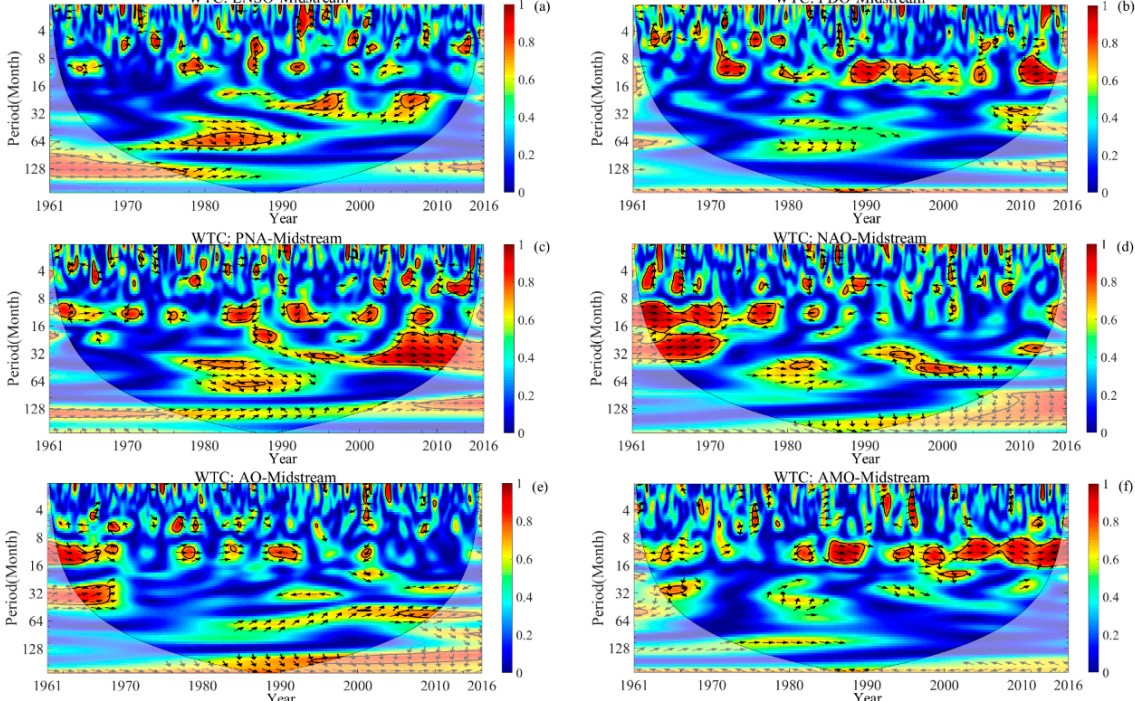

**Figure 9.** The wavelet coherence spectra between monthly observed average precipitation and El Niño-Southern Oscillation (ENSO) (**a**), Pacific Decadal Oscillation (PDO) (**b**), Pacific North American teleconnection pattern (PNA) (**c**), North Arctic Oscillation (NAO) (**d**), Atlantic Oscillation (AO) (**e**), and Atlantic Multidecadal Oscillation (AMO) (**f**) indices covering 1961–2016 in the middle Heihe River Basin (MHRB). The 5% significance level against red noise is exhibited as a thick contour, and the relative phase relationship is denoted by arrows.

**Table 4.** Correlations between the large-scale climate indices and lag-time coefficients of precipitation in the upper (UHRB) and middle (MHRB) reaches of the Heihe River Basin.

| Regions | Lag Time (Months) | Correlations | ENSO | AO | NAO | PNA | PDO | AMO |
|---|---|---|---|---|---|---|---|---|
| UHRB | 0 | Pearson | −0.016 | 0.033 | 0.077 [a] | 0.019 | −0.020 | 0.168 [b] |
| | | Kendall | 0.009 | 0.024 | 0.072 [b] | −0.016 | −0.014 | 0.082 [b] |
| | | Spearman | 0.014 | 0.041 | 0.106 [b] | −0.025 | −0.021 | 0.122 [b] |
| | 1 | Pearson | −0.016 | 0.052 | 0.053 | −0.014 | 0.039 | 0.162 [b] |
| | | Kendall | 0.012 | 0.040 | 0.064 [a] | −0.024 | 0.033 | 0.087 [b] |
| | | Spearman | 0.019 | 0.065 | 0.094 [a] | −0.034 | 0.050 | 0.130 [b] |
| | 3 | Pearson | −0.003 | 0.063 | 0.014 | −0.058 | 0.145 [b] | 0.084 |
| | | Kendall | 0.015 | 0.022 | −0.001 | −0.021 | 0.095 [b] | 0.056 [a] |
| | | Spearman | 0.022 | 0.031 | −0.001 | −0.030 | 0.14 [b] | 0.081 [a] |
| | 6 | Pearson | 0.006 | −0.042 | −0.094 [a] | 0.044 | 0.060 | −0.065 |
| | | Kendall | −0.009 | 0.001 | −0.061 [a] | 0.017 | 0.035 | −0.038 |
| | | Spearman | −0.015 | −0.007 | −0.090 [a] | 0.027 | 0.053 | −0.056 |
| | 9 | Pearson | 0.047 | −0.019 | 0.015 | 0.027 | −0.106 [b] | −0.036 |
| | | Kendall | 0.002 | −0.029 | −0.002 | 0.038 | −0.070 [b] | −0.021 |
| | | Spearman | 0.004 | −0.043 | −0.004 | 0.055 | −0.103 [b] | −0.030 |
| | 12 | Pearson | 0.030 | 0.013 | 0.058 | 0.023 | 0.006 | 0.143 [b] |
| | | Kendall | 0.019 | 0.004 | 0.068 [b] | −0.016 | −0.005 | 0.076 [b] |
| | | Spearman | 0.027 | 0.012 | 0.100 [b] | −0.024 | −0.007 | 0.113 [b] |
| MHRB | 0 | Pearson | −0.013 | 0.048 | 0.032 | 0.027 | 0.024 | 0.112 [b] |
| | | Kendall | 0.027 | 0.039 | 0.056 [a] | 0.010 | 0.010 | 0.090 [b] |
| | | Spearman | 0.044 | 0.069 | 0.085 [a] | 0.018 | 0.015 | 0.133 [b] |
| | 1 | Pearson | −0.003 | 0.059 | 0.037 | −0.025 | 0.099 [a] | 0.104 [b] |
| | | Kendall | 0.036 | 0.020 | 0.015 | −0.014 | 0.053 [a] | 0.093 [b] |
| | | Spearman | 0.059 | 0.029 | 0.022 | -.022 | 0.079 [a] | 0.139 [b] |
| | 3 | Pearson | 0.040 | 0.031 | −0.019 | −0.005 | 0.153 [b] | 0.032 |
| | | Kendall | 0.050 | 0.019 | −0.024 | 0.001 | 0.101 [b] | 0.043 |
| | | Spearman | 0.076 [a] | 0.030 | −0.036 | 0.003 | 0.148 [b] | 0.066 |
| | 6 | Pearson | 0.067 | −0.059 | −0.070 | 0.051 | 0 | −0.076 |
| | | Kendall | 0.038 | −0.021 | −0.063 [a] | 0.038 | 0.013 | −0.031 |
| | | Spearman | 0.056 | −0.033 | −0.092 [a] | 0.058 | 0.020 | −0.046 |
| | 9 | Pearson | 0.090 [a] | 0.016 | 0.026 | −0.007 | −0.063 | 0.001 |
| | | Kendall | 0.050 | 0.008 | 0.027 | 0.035 | −0.037 | 0.004 |
| | | Spearman | 0.075 | 0.011 | 0.041 | 0.051 | −0.055 | 0.007 |
| | 12 | Pearson | 0.038 | 0.042 | 0.066 | 0.015 | 0.028 | 0.087 [a] |
| | | Kendall | 0.034 | 0.026 | 0.063 [a] | −0.015 | 0.011 | 0.065 [a] |
| | | Spearman | 0.049 | 0.045 | 0.094 [a] | −0.022 | 0.017 | 0.096 [a] |

[a] Significant at the 0.05 level. [b] Significant at the 0.01 level.

Li et al. [37] and Li et al. [38] reported that there were strong significant associations between the annual precipitation and the WPSH and the NASH in northwestern China and Gansu Province (Pearson correlation coefficients). They also pointed out that the significant increasing trend in precipitation in northwest China most likely reflected the strengthening of the WPSH and NASH, observed since the mid-1980s. Li et al. [38] demonstrated that the relationships between large-scale climate indices and precipitation indices were less severe in northwestern Gansu. However, we found that there were significant correlations between the monthly climate indices and precipitation using the wavelet coherence spectra analysis method, and significant lagged correlations between precipitation and the NAO, PDO, and AMO indices using the Pearson, Kendall, and Spearman correlation coefficients. Therefore, the different results may be related to different research methods and data time scales. For example, Mariotti et al. [67] and Sun et al. [68] reported that different simulation methods might lead to different research results in the same study areas, particularly for precipitation. Li et al. [37] also pointed out that water vapor in the arid region of northwest China may originate from a range of places, such as the western Atlantic, northern Eurasia, the Arctic Ocean, the eastern Pacific Ocean, and

the southern Indian Ocean. Thus, the correlations between climate indices and precipitation need to be examined in more detail in the future.

### 4.7. Relationship between Precipitation and the NDVI

The variations in the annual and summer NDVI in the UHRB and MHRB from 1998 to 2015 are shown in Figure 10. The annual and summer NDVI significantly increased at rates of 0.0025/a and 0.0032/a, respectively. These results are consistent with those reported by Sun et al. [27] and Guan et al. [35], who both found that changes in vegetation growth in the Qilian Mountains may have been related to increases in precipitation. Guan et al. [35] also found that the NDVI was negatively correlated with precipitation in oasis areas. Cultivated vegetation, coniferous forests, broad-leaved forests, and grasslands showed increased NDVI, with the greatest increase in cultivated vegetation, while alpine vegetation decreased (Figures 1c and 11a), but none of the changes were significant ($t < 2.120$). The NDVI of grasslands, meadows and coniferous forests were significantly and positively correlated ($p < 0.01$), but the NDVI of, alpine vegetation, swamps and shrubs were negatively and significantly correlated ($p < 0.01$) in the UHRB. The NDVI of grasslands was significantly and positively correlated ($p < 0.01$), but the NDVI of shrubs, coniferous forests and cultivated vegetation were negatively and significantly correlated ($p < 0.01$) in the MHRB (except for cultivated vegetation in some grain for green areas showed a positive correlation). This is consistent with the results of Zhou et al. [24], who found that grasslands in the UHRB and MHRB were positively correlated with precipitation ($r = 0.922$, $p < 0.05$). Also, the fact that cultivated vegetation (oasis areas) was significantly and negatively correlated with precipitation agrees with Guan et al. [35]. These results, however, are inconsistent with those of Chuai et al. [25] and Zhong et al. [69]. In addition, Han et al. [70] found that temperature and precipitation were important influences on vegetation changes in the HRB, and that periodic change in climatic factors was mainly affected by atmospheric movement, with consequences for vegetation. They also reported that natural precipitation did not directly control vegetation growth in these areas, and that water resources could be used efficiently for irrigation, for example, through water-saving techniques like low-pressure pipe irrigation, sprinkler irrigation, drip irrigation, and narrow border irrigation. Irrigation has, therefore, reduced the relationship of NDVI with natural precipitation [23] and, through rational and effective management of the oasis, it has been possible to improve the local vegetation [35]. In addition, Xu et al. [71] pointed out that there were 24 irrigation districts, with thousands of canals and over 6000 pumping wells in the MHRB, which alleviate the shortages of natural precipitation. Total irrigation water demand in the MHRB had increased by $3.249 \times 10^8$ m$^3$ over the period 1985–2014 [72]. This may help to explain the significant negative correlation between cultivated vegetation and precipitation (Figure 11b).

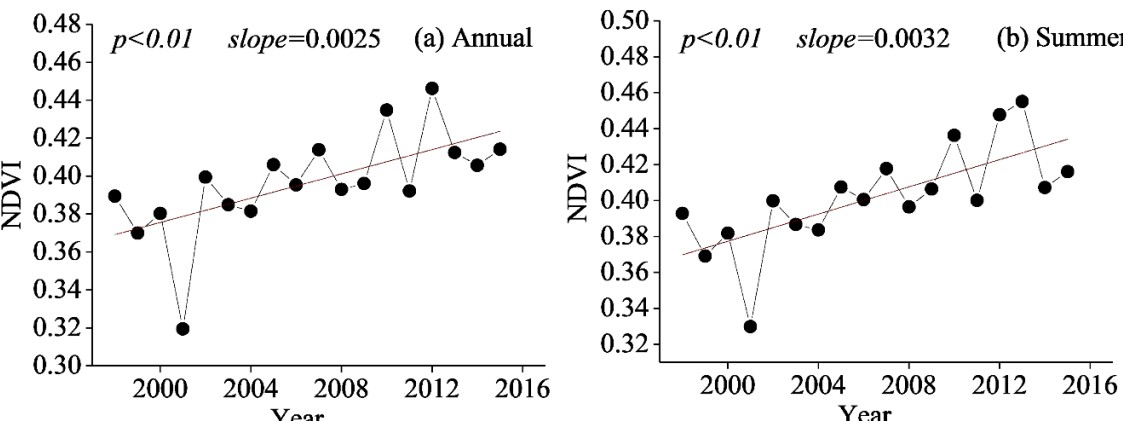

**Figure 10.** The change trend of the annual and summer (June-July-August) NDVI in the upper (UHRB) and middle (MHRB) reaches of the Heihe River Basin 1998–2015.

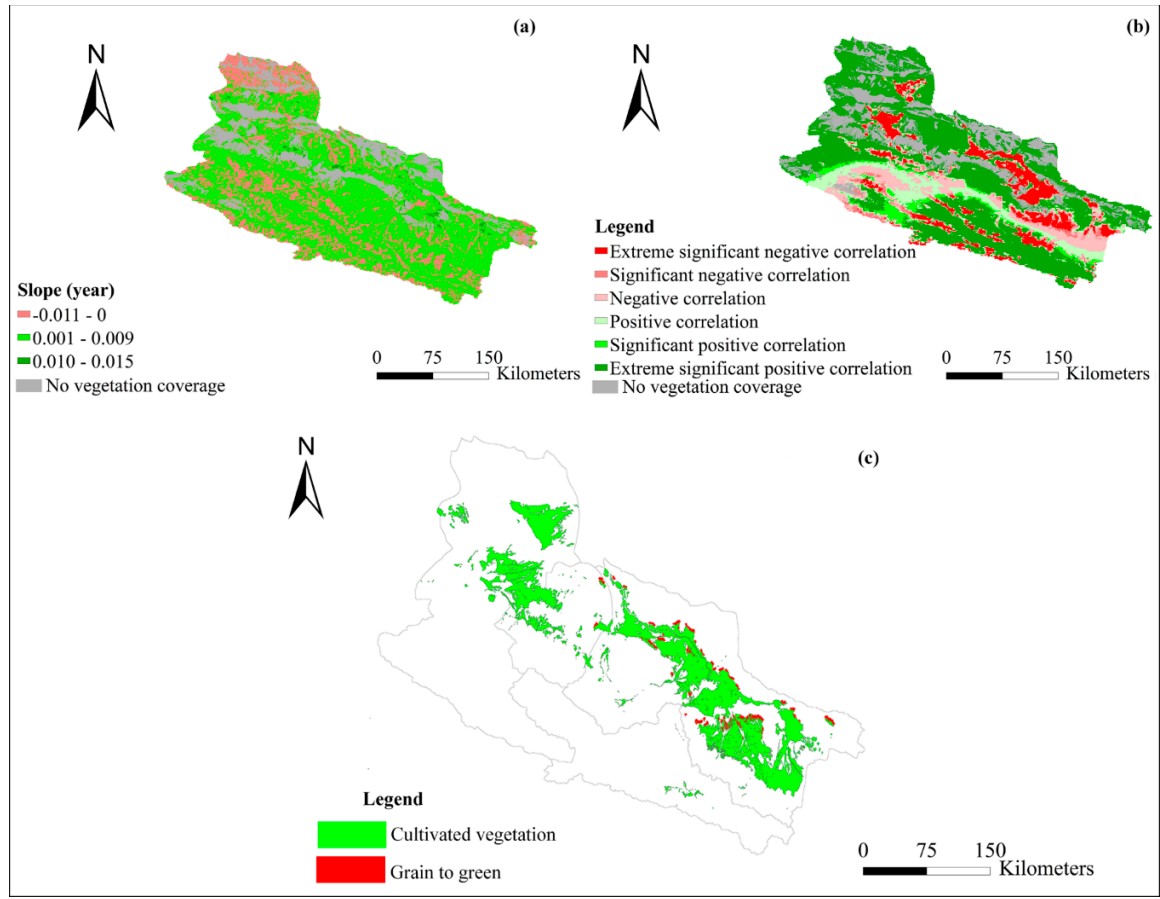

**Figure 11.** The annual change trend of the normalized difference vegetation index (NDVI) (**a**), and correlations of the NDVI precipitation (**b**), and grain for green (**c**) in the upper (UHRB) and middle (MHRB) reaches of the Heihe River Basin 1998–2015.

LUCC can have considerable effects on the NDVI [73] and the NDVI is regarded as an indicator of LUCC [74,75]. We found that the NDVI and precipitation correlation shows strong dependence on land cover type and vegetation types (Figure 1b,c and Figure 11b,c), while from Figure 12 and Table 5, that cultivated land and built land had increased, and grasslands and unused land had decreased, especially since 2000 (Ecological Water Diversion Project), and that the land use changes were most obvious in the MHRB from 1975 to 2010. For example, conversions from grassland ($3.97 \times 10^4$ ha) and unused land ($1.84 \times 10^4$ ha) to cultivated land accounted for most land use changes. The NDVI increased in the UHRB and MHRB between 1998 and 2015, and the relationship between the NDVI and precipitation was significant and positive in the UHRB (grassland and coniferous forest) and significant and negative in the MHRB (cultivated vegetation). Liu et al. [76] also pointed out that human agricultural activities were the dominant forces driving the increase in water requirements. The contribution of oasis expansion to the increased water needs was significantly greater than that of other variables. This reveals that controlling the oasis scale is extremely important and effective for balancing water for agriculture and ecosystems and for achieving sustainable oasis development in arid regions. Hence, these findings suggest that, because of human activities, the correlation between cultivated vegetation and natural precipitation in the MHRB may have weakened.

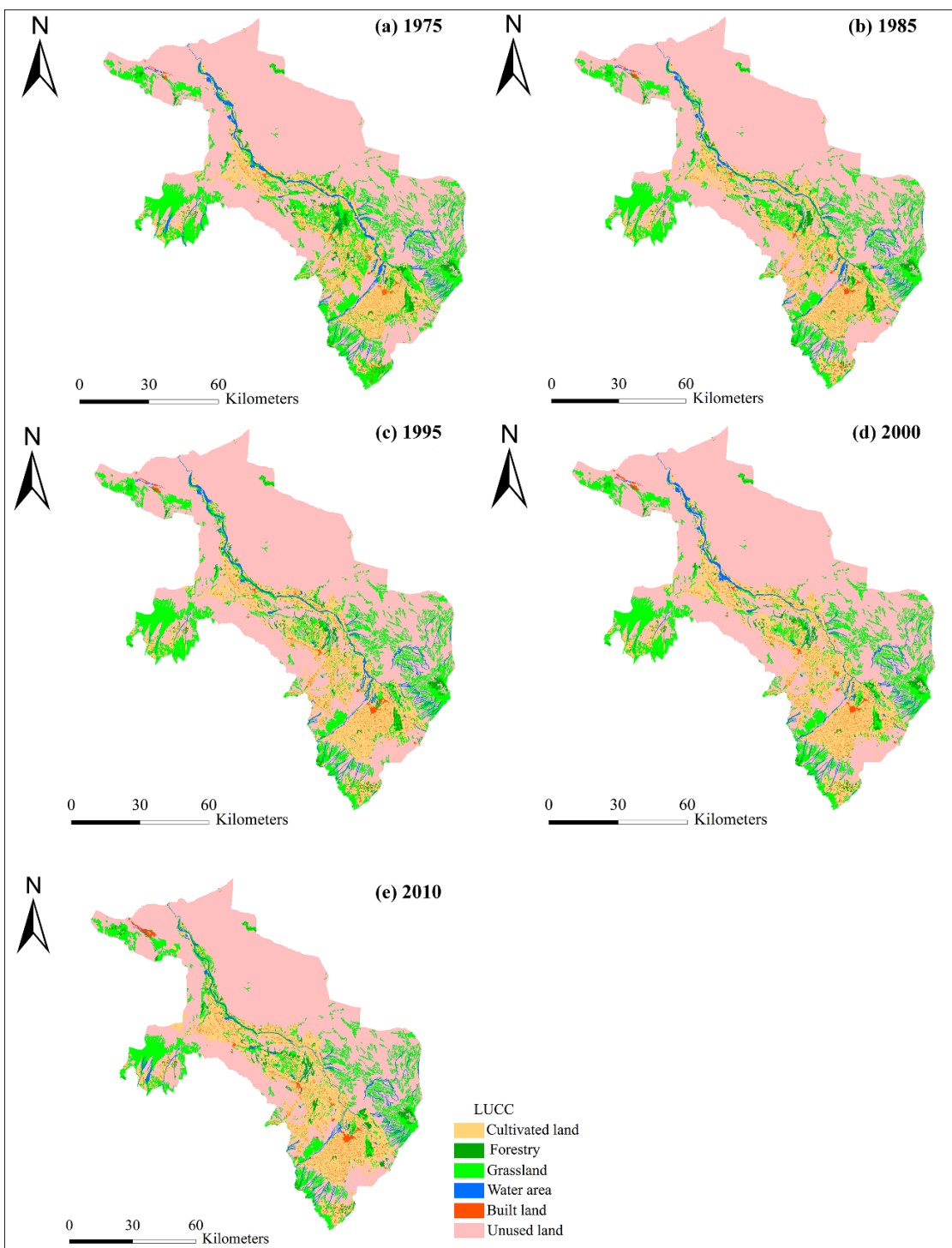

**Figure 12.** Spatial distribution of the changes in land use/cover pattern in the middle Heihe River Basin (MHRB) 1975–2010.

Changes in precipitation can be central to quantifying the impact of climate change on the hydrological cycle. Also, changes in precipitation can mirror the variations of the hydrological cycle on a regional or global scale. Spatial and temporal variations of precipitation are, therefore, vital for determining the spatial and temporal distributions of water resources [12].

Furthermore, precipitation, as the only source of water in the inland watershed of an arid area, not only determines the types and spatial distribution of natural ecosystems, but also plays

an important role in agricultural yields, economic development, and societal status [77]. Studies of inland watersheds of arid areas are therefore very valuable [78,79]. In this study, we analyzed the changes in the precipitation patterns and how they were correlated with climate indices and the NDVI. We obtained some valuable insights into the changes in precipitation in the HRB and how they were correlated with trends in vegetation. However, any analysis of changes in precipitation is subject to many influences, such as the research method, data timescale, number of meteorological stations, and the spatial resolution of the remote sensing dataset, all of which may introduce uncertainty.

**Table 5.** Transition matrix and the primary types of land use changes in the middle Heihe River Basin (MHRB) 1975–2010 ($10^4$ ha$^2$).

| Land Use Types | Cultivated Land | Woodland | Grassland | Water Area | Built on Land | Unused Land |
|---|---|---|---|---|---|---|
| Cultivated land | | 0.04 | 0.26 | 0.05 | 0.28 | 0.15 |
| Woodland | 0.42 | | 0.21 | 0.01 | 0.01 | 0.12 |
| Grassland | 3.97 | 0.26 | | 0.16 | 0.06 | 1.71 |
| Water area | 0.53 | 0.02 | 0.34 | | 0.01 | 0.19 |
| Built land | 0.06 | 0.00 | 0.00 | 0.01 | | 0.00 |
| Unused land | 1.84 | 0.38 | 1.46 | 0.14 | 0.22 | |
| | **Cultivated Land** | **Woodland** | **Grassland** | **Water Area** | **Built on Land** | **Unused Land** |
| 1975–1987 | 14.40 | 1.52 | 15.33 | 2.22 | 1.60 | 65.52 |
| 1987–1992 | 15.25 | 1.21 | 14.49 | 2.04 | 1.34 | 66.88 |
| 1992–2001 | 16.48 | 1.23 | 14.29 | 2.11 | 1.41 | 67.04 |
| 2000–2010 | 17.88 | 1.25 | 13.03 | 1.80 | 1.71 | 63.87 |

## 5. Conclusions

Spatial and temporal variations of the occurrence and fractional contributions of WPs to the total precipitation amount (days), precipitation indices and the linkages with climatic indices were analyzed in the UHRB and MHRB. The precipitation episodes in the UHRB and MHRB were dominated by the occurrence and fractional contribution of WPs with durations of 1–4 days and 1–2 days, respectively. The occurrences and fractional contribution of longer-duration WPs decreased from the 1990s until 2016, while those of shorter WPs increased in the UHRB and MHRB. WPs occurred more frequently in the UHRB than in the MHRB, but the precipitation concentration was higher in the MHRB (concentration 1 day) than in the UHRB. Thus, the risk of flash floods may be higher in the UHRB than in the MHRB, while urban waterlogging (over 90% of the population and major cities are concentrated in the MHRB) and drought risk may have grown in the MHRB. The spatial distribution of the fractional contributions and occurrences were almost consistent with the stations where the precipitation was increasing or decreasing.

The results of this analysis showed that the ATP, ATD, API, and AMRD increased in the UHRB and MHRB over the study period. Increases in the ATP and ATD caused an increase in the API in the UHRB and MHRB, while the AMRD also increased. The dry seasons became slightly wetter, and the wet seasons became slightly drier, but the precipitation regimes did not shift.

Precipitation and climate indices showed significant but variable correlations for ENSO, AO, NAO, PDO, PNA and AMO, especially for PDO, NAO and AMO at 4- to 78.6-month timescales. The results from Pearson, Kendall, and Spearman tests showed that there were weak lagged linkages between precipitation and the PDO (lag of 3 months), and AMO (lag of 1 month) indices in the UHRB, and NAO (lag of 12 months), PDO (lag of 3 months), and AMO (lag of 1 month) in the MHRB.

The NDVI increased significantly in the UHRB and MHRB from 1998 to 2015. It was significantly and positively correlated with the NDVI of grasslands, meadows, and coniferous forests. The NDVI of alpine vegetation, swamps, and shrubs were negatively and significantly correlated with precipitation in the UHRB. The NDVI of grasslands was significantly and positively correlated, but the NDVI of shrubs, coniferous forests, and cultivated vegetation were negatively and significantly correlated ($p < 0.01$) with precipitation in the MHRB. Human activities may have weakened the correlation between cultivated vegetation and natural precipitation in the MHRB.

These results point to new challenges for water supply and for water resources management under the impact of climate change and human activities, therefore, collaborative and sustainable water resources allocations are needed in the UHRB and MHRB.

**Author Contributions:** Conceptualization, Methodology and Data curation, F.Z., Q.C.; Figures and Tables, Q.C.; writing—original draft preparation, F.Z., Q.C.; writing—review and editing, F.Z., Y.G.

**Funding:** This research was funded by the Strategic Priority Research Program of the Chinese Academy of Sciences (Grant No. XDA19040500), National Natural Science Foundation of China (Grant No. 41571516, 41471448), and the Fundamental Research Funds for the Central Universities (Grant No.2019jbkyjd013).

**Conflicts of Interest:** The authors declare no conflict of interest.

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
