# Peer review of "Relationships between Spatial and Temporal Variations in Precipitation, Climatic Indices, and the Normalized Differential Vegetation Index in the Upper and Middle Reaches of the Heihe River Basin, Northwest China"

_water, doi:10.3390/w11071394_

Round 1
Reviewer 1 Report
The main issue in this manuscript is that the correlation coefficients between climate indices and precipitation are too low ( table 4), can't support the conclusion well. Please see the following comments for details.
Page 7, section 4.2 "Figure 3 shows the temporal patterns in the normalized occurrence and fractional contribution from the meteorological stations over the UHRB and MHRB."
What's the normalized occurrence and fractional contribution? need to explain clearly so that people can understand the figure 3.
In addition, need to revise the caption of figure 3 to indicate a), b), c) and d) clearly.
Page 7, line 238-239. "These results contrast with the findings of Zolina et al. [16] from
their study in Europe, ..."
Perhaps use other word instead of "contrast", the results may vary with regions.
Page 7, line 246-248. "Hence, the occurrences and fractional contributions of
WPs of longer durations decreased, whereas shorter durations increased,
especially after the 1990s, which means that shorter WPs occurred more
frequently in the UHRB and MHRB." How about the 1960s? It is OK to say
decrease since 1970s, but the results of 1960s are different.
Page 8-9, section 4.3. Line 283-284. "..., and that the increases and decreases in the fractional contributions and occurrences were mostly non-significant at all stations in the UHRB and MHRB". What does this mean, no significant increase/decrease changes found? If so, consistent with the results in section 4.2?
Page 10, section 4.4, page 10-12. Based on table 3, trends of precipitation indices at most stations are not statistically significant.
Page 15, table 4. The most of the correlation coefficients are so small, and not statistically significant. So, can't draw meaningful conclusion based on these results.
Page 16, section 4.7. What's the period of summer in this study? Based on figure 10, the annual NDVI values are close to summer NDVI values, and some of the annual NDVI values are even higher than summer NDVI values of the same year. The results are not reasonable.
Page 18, line 488-490. "Precipitation and climate indices were significantly correlated. The results from Pearson, Kendall, and Spearman tests showed that there were significant lagged correlations between precipitation and the NAO, PDO, and AMO indices in the UHRB and MHRB." Unfortunately, the results in table 4 can't support this conclusion. Most of the correlation coefficients are too small to mean anything, and not statistically significant.
Author Response
Response to the reviewer’ comments
We would like to thank the anonymous reviewer for reviewing our manuscript. We especially appreciate their constructive comments. We agree that some aspects of the paper needed further improvement and extension in order to make it more generally valid and suitable for publication. In the following, we address all comments point-by-point in more detail according to reviewers’ comments. All revisions are highlighted in the context.
Reviewer 1
Comments and Suggestions for Authors
The main issue in this manuscript is that the correlation coefficients between climate indices and precipitation are too low (table 4), can't support the conclusion well. Please see the following comments for details.
-Answer: Thanks a lot for the comments. Changes in precipitation are critical indicators of climate change. Climate change has a relatively large impact on various aspects of precipitation including total precipitation, precipitation intensity, and the number of rainy days [4]. Precipitation is one of the most important components of the hydrological cycle. Human activities influence water resource management, with effects on natural disasters (e.g., drought and flood), agricultural productivity, economic development, and the ecological environment [1,5-6]. While water availability and temperature both limit vegetation growth, water availability is the more important. Precipitation, therefore, plays an important role in the geographical distribution of vegetation [19-21]. Climate change (i.e., changes in temperature and precipitation) and human activities also affect changes in the normalized differential vegetation index (NDVI). Additionally, Li et al. [38], Yang et al. [6], and Qin et al. [43] all reported that significant increases in precipitation were closely associated with the West Pacific Subtropical High (WPSH) and the North America Subtropical High (NASH). The relationship between precipitation and the El Niño-Southern Oscillation (ENSO) varied between different periods and on different time scales [6], and seemed to oscillate over periods of 3, 6, and 11 years [43] in Northwest China. To the best of our knowledge, the relationships between precipitation indices and large-scale circulation patterns and WPs have not yet been established for the Heihe River Basin. Therefore, the Pearson, Kendall, and Spearman correlation coefficients; Theil-Sen Median; Mann–Kendall test, and wavelet coherence are used in this study. The results indicated that the occurrences (fractional contributions) of 1–2-day wet periods were 81.3% (93.9%) and 55.3% (82.1%) in the upper (UHRB) and middle (MHRB) reaches of the Heihe River Basin, respectively. The spatial distribution of the occurrences was almost consistent with non-significant increases/decreases at stations. The ATP, ATD, API, and AMRD increased, while precipitation regimes suggest that dry seasons were getting wetter, and wet seasons were getting drier, but these changes were not significant. Wavelet coherence analyses showed that climate indices influenced precipitation, mainly its concentration, on a 4- to 78.6-month timescale. The Pearson, Kendall, and Spearman correlation coefficients showed significant lagged correlations between precipitation and the North Arctic Oscillation (NAO), Pacific Decadal Oscillation (PDO), and Atlantic Multidecadal Oscillation (AMO). The NDVI was significantly and positively correlated with precipitation in the UHRB and significantly and negatively correlated with precipitation in the MHRB. The correlation between cultivated vegetation and natural precipitation in the MHRB may have been weakened by human activities.
We agree that some aspects of the paper needed further improvement. Correlations are low, low correlation coefficients with significant p-values may occur due to the large sample size used in this study, which resulted in high degrees of freedom [66]. Furthermore, De Oliveirajunior et al [67] revealed that all tests, regardless of the Standardized Precipitation Index scale, showed low Pearson, Kendall, and Spearman correlation coefficients, while significant for ENSO and PDO in the North and Northwest regions, State of Rio de Janeiro-Brazil. It is concluded that the ENSO and PDO signals are unclear and, therefore, do not influence precipitation independently. The analysis demonstrated that low significance Pearson, Kendall, and Spearman lag correlation coefficients for precipitation and the climate indices are possibly attributed to high degrees of freedom and that climate indices do not affect precipitation independently.
Page 7, section 4.2 "Figure 3 shows the temporal patterns in the normalized occurrence and fractional contribution from the meteorological stations over the UHRB and MHRB."
What's the normalized occurrence and fractional contribution? need to explain clearly so that people can understand the figure 3.
In addition, need to revise the caption of figure 3 to indicate a), b), c) and d) clearly.
Answer: Thanks a lot for pointing this out. According to the comments, we added more information in the section 4.2 (P7 L238-240, P7 L259-261). For example:
Figure 3 shows the temporal patterns in the normalized occurrence and fractional contribution. The normalized occurrence and fractional contribution were derived as Z= (X−`X)/Std(x), where Z is the normalized series, X represent occurrence and fractional precipitation series,`X is the average value of X, and Std(X) denotes the standard deviation of X. All the normalized scores were smoothed by a 5-year running average obtained from the meteorological stations over the UHRB and MHRB.
…………………………………
Figure 3. Temporal evolution of normalized occurrences (a denotes UHRB, b denotes MHRB) and fractional contribution (c denotes UHRB, d denotes MHRB) anomalies in different WPs in the UHRB and MHRB.
……………………………
Page 7, line 238-239. "These results contrast with the findings of Zolina et al. [16] from
their study in Europe, ..." Perhaps use other word instead of "contrast", the results may vary with regions.
-Answer: Thanks a lot for pointing this out. We revised this part according to the comments (P7 L244-245). For example:
These results were different from the findings of Zolina et al. [16] in their study in Europe, but, although the duration days differed on the interdecadal scale,………….
Page 7, line 246-248. "Hence, the occurrences and fractional contributions of WPs of longer durations decreased, whereas shorter durations increased, especially after the 1990s, which means that shorter WPs occurred more frequently in the UHRB and MHRB." How about the 1960s? It is OK to say decrease since 1970s, but the results of 1960s are different.
-Answer: Thanks a lot for pointing this out. We revised this part according to the comments (P7 L241-242, L248-250). For example:
…………………The temporal pattern in the fractional contributions of WPs was similar to that of the occurrence of the WPs. Figure 3c shows that 5–6-day WPs during the 1960s and 7–8-day WPs during the 1970s and 1980s made the greatest fractional contributions to the annual total precipitation……………………….
Page 8-9, section 4.3. Line 283-284. "..., and that the increases and decreases in the fractional contributions and occurrences were mostly non-significant at all stations in the UHRB and MHRB". What does this mean, no significant increase/decrease changes found? If so, consistent with the results in section 4.2?
-Answer: Thanks a lot for pointing this out. We are sorry that this part is unclear.
Section 4.2 mainly analyses the temporal trend for the whole upstream and middle reaches. section 4.3 analyses the spatial characteristics of 10 meteorological stations for before and after an abrupt change point, total period (1961–2016).
We revised it and added the relevant information in section 4.3 (P8-9 L263-266, L290-293). For example:
Figure 4 and 5 show the Mann–Kendall trend for the Z value of the spatial distribution of normalized occurrences and fractional contributions of WPs with different durations, i.e., 1–2 days; 3–5 days in 10 stations before and after an abrupt change point, total period (1961–2016). The significance of trends is evaluated at the 0.01 level…………………
……………The above analysis shows that the spatial distribution of occurrences and fractional contributions were almost consistent with the spatial distribution patterns, and that the significant increases and decreases for the fractional contributions and occurrences were sporadic for some stations of WPs (i.e., 1-day, 2-days) in the UHRB and MHRB……………….
Page 10, section 4.4, page 10-12. Based on table 3, trends of precipitation indices at most stations are not statistically significant.
-Answer: Thanks a lot for pointing this out. Yes, based on table 3, trends of precipitation indices at most stations are not statistically significant. However, As shown in table 2, the ATP, ATD, API and AMRD increased at all stations except the ATD in Menyuan, the ATD in Gaotai, and the AMRD in Gangcha. The ATP increases were significant at 5 stations. The ATD increases were statistically significant at 3 stations (Table 2). The API and AMRD had significant increases at 2 and 3 stations, respectively. We revised this part according to the comments (P10 L307-310).
Page 15, table 4. The most of the correlation coefficients are so small, and not statistically significant. So, can't draw meaningful conclusion based on these results.
-Answer: Thank you for your comment. We agree with the correlation coefficients are low. Correlations are low, low correlation coefficients with significant p-values may occur due to the large sample size used in this study, which resulted in high degrees of freedom [66]. Furthermore, De Oliveirajunior et al [67] revealed that all tests, regardless of the Standardized Precipitation Index scale, showed low Pearson, Kendall, and Spearman correlation coefficients, while significant for ENSO and PDO in the North and Northwest regions, State of Rio de Janeiro-Brazil. It is concluded that the ENSO and PDO signals are unclear and, therefore, do not influence precipitation independently. The analysis demonstrated that low significance Pearson, Kendall, and Spearman lag correlation coefficients for precipitation and the climate indices are possibly attributed to high degrees of freedom and that climate indices do not affect precipitation independently. For example (P16 L400-419)
Page 16, section 4.7. What's the period of summer in this study? Based on figure 10, the annual NDVI values are close to summer NDVI values, and some of the annual NDVI values are even higher than summer NDVI values of the same year. The results are not reasonable.
-Answer: Thanks for pointing this out. Here, the summer period is June-July-August. After careful checking, we found that this error arose from the Origin software, when the order of the combination of pictures was reversed. This is a very low-level mistake, which gives us a great warning that it is necessary to examine manuscripts carefully and repeatedly between submissions. We revised it (P16 L441).
Page 18, line 488-490. "Precipitation and climate indices were significantly correlated. The results from Pearson, Kendall, and Spearman tests showed that there were significant lagged correlations between precipitation and the NAO, PDO, and AMO indices in the UHRB and MHRB." Unfortunately, the results in table 4 can't support this conclusion. Most of the correlation coefficients are too small to mean anything, and not statistically significant.
-Answer: Thank you for your comment. We agree with the correlation coefficients are low.
The analysis demonstrated that low significance Pearson, Kendall, and Spearman lag correlation coefficients for precipitation and the climate indices are possibly attributed to high degrees of freedom and that climate indices do not affect precipitation independently. We revised this part (P10-11 L400-419). For example:
………………However, when correlations are low, low correlation coefficients with significant p-values may occur due to the large sample size used in this study, which resulted in high degrees of freedom [66]. Furthermore, De Oliveirajunior et al [67] revealed that all tests, regardless of the Standardized Precipitation Index scale, showed low Pearson, Kendall, and Spearman correlation coefficients, while significant for ENSO and PDO in the North and Northwest regions, State of Rio de Janeiro-Brazil. It is concluded that the ENSO and PDO signals are unclear and, therefore, do not influence precipitation independently. The analysis demonstrated that low significance Pearson, Kendall, and Spearman lag correlation coefficients for precipitation and the climate indices are possibly attributed to high degrees of freedom and that climate indices do not affect precipitation independently………………….
References
[1] Gao: Y.; He, N.; Wang, Q.; Miao, C.Y. Increase of external nutrient input impact on carbon sinks in Chinese coastal seas. Environ. Sci. Technol. 2013,47,13215–13216. http://dx.doi.org/10.1021/es4045743.
[4] Yu, H.; Wang, L.; Yang, R.; Yang, M.L.; Gao, R. Temporal and spatial variation of precipitation in the Hengduan mountains region in China and its relationship with elevation and latitude. Atmos. Res. 2018,213,1-16. http://dx.doi.org/10.1016/j.atmosres.2018.05.025.
[5] Sayemuzzaman, M.; Jha, M.K. Seasonal and annual precipitation time series trend analysis in North Carolina, United States. Atmos. Res. 2014,137,183–194. http://dx.doi.org/10.1016/j.atmosres.2013.10.012.
[6] Yang, P.; Xia, J.; Zhang, Y.Y.; Hong, S. Temporal and spatial variations of precipitation in northwest china during 1960–2013. Atmos. Res. 2017,183,283-295. http://dx.doi.org/10.1016/j.atmosres.2016.09.014.
[11] Zhang, Q.; Singh, V.P.; Peng, J.; Chen, Y.D.; Li, J.F. Spatial–temporal changes of precipitation structure across the pearl river basin, China. J. Hydrol. 2012, 440-441(03),113-122. http://dx.doi.org/10.1016/j.jhydrol.2012.03.037.
[12] Zhang, Q.; Singh, V.P.; Li, J.F.; Chen, X.H. Analysis of the periods of maximum consecutive wet days in China. J. Geophys. Res Atmos. 2011, 116(D23),1-18. http://dx.doi.org/10.1029/2011JD016088.
[16] Zolina, O.; Simmer, C.; Gulev, S.; Kollet, K.S. Changing structure of European precipitation: Longer wet periods leading to more abundant rainfalls. Geophys. Res. Lett. 2010, 37, L06704. http://dx.doi.org/doi:10.1029/2010GL042468.
[17] Wang, Y.; Zhang, Q.; Singh, V.P. Spatiotemporal patterns of precipitation regimes in the Huai river basin, China, and possible relations with ENSO events. Nat. Hazards. 2016,82(3),1-19. http://dx.doi.org/10.1007/s11069-016-2303-3.
[18] Huang, J.; Chen, X.; Xue, Y.; Lin, J.; Zhang, J. Changing characteristics of wet/dry spells during 1961–2008 in Sichuan province, southwest China. Theor. Appl. Climatol. 2017,127(1-2), 129-141. http://dx.doi.org/10.1007/s00704-015-1621-9.
[19] Piao, S.L.; Fang, J.Y.; Ji,W.; Guo, Q.H.; Ke, J.H.; Tao, S. Variation in a satellite-based vegetation index in relation to climate in China. J. Veg. Sci. 2004,15,219–226. http://dx.doi.org/10.1658/1100-9233(2004)015[0219:VIASVI]2.0.CO;2.
[20] Cleland, E.E.; Chuine, I.; Menzel, A.; Mooney, H.A.; Schwartz, M.D. Shifting plant phenology in response to global change. Trends.Ecol. Evol. 2007, 22(7),357–365. http://dx.doi.org/10.1016/j.tree.2007.04.003.
[21] Mao, D.H.; Wang, Z.M.; Luo L.; Ren, C.Y. Integrating AVHRR and MODIS data to monitor NDVI changes and their relationships with climatic parameters in Northeast China. Int. J. Appl. Earth. Obs. Geoinf. 2012,18(1),528–536. http://dx.doi.org/10.1016/j.jag.2011.10.007.
[38] Li, B.F., Chen, Y.N., Chen, Z.S., Xiong, H.G., Lian, LS. Why does precipitation in northwest China show a significant increasing trend from 1960 to 2010. Atmos. Res. 2016, 275-284. http://dx.doi. org/10.1016/j.atmosres.2015.08.017.
[43] Qin, Y.H.; Li, B.F.; Chen, Z.S.; Chen, Y.N.; Lian, L.S. Spatio‐temporal variations of nonlinear trends of precipitation over an arid region of northwest China according to the extreme‐point symmetric mode decomposition method. Int. J. Climatol. 2017,38(11),1-11. http://dx.doi.org/10.1002/joc.5330
[66] Nalley, D.; Adamowski, J.; Khalil, B.; Biswas, A. Inter-annual to inter-decadal streamflow variability in Quebec and Ontario in relation to dominant largescale climate indices. J. Hydrol. 2016,536,426–446. http://dx.doi.org/10.1016/j.jhydrol.2016.02.049.
[67] De Oliveirajunior, J. F.; Gois, G.; Terassi, P. M.; Junior, C. A.; Blanco, C. J.; Sobral, B. S.; Gasparini, K. A. Drought severity based on the SPI index and its relation to the ENSO and PDO climatic variability modes in the regions North and Northwest of the State of Rio de Janeiro-Brazil. Atmos. Res. 2018, 212,91-105. http://dx.doi.org/10.1016/j.atmosres.2018.04.022.

Reviewer 2 Report
- Comments : At the first I would like to thank authors for very interesting work and hard work. This study examined temporal and spatial variations in the precipitation indices, and analyzed the relationships between the monthly precipitation and large-scale circulations in the upper and middle reaches of Heihe river basin, northwest China. The study also analyzed the relationship between precipitation and the NDVI and suggested that human activity could weaken the correlation between vegetation and natural precipitation. The results of this study will help to analyze the precipitation indices, and the relationship between precipitation and large-scale circulations / vegetation. Below are the comments, which I believe, can improve the work. #1. Table 2: Chapter 2 describes the study area and Table 2 does not seem to match the contents of this chapter. It may be necessary to move to another chapter or add a new chapter. #2. Chapter 3.2.3: “Pearson, Kendall and Spearman tests” analyze correlations, so 'Correlation tests (Pearson, Kendall and Spearman)' would be better than 'Other Methods' for chapter titles (It would be nice if it would be modified to include all three.). Also, it seems that there is a lack of explanation compared to the other methods above. It would be nice to add a description. #3. Figure 3: The color bars in the four figures (a) to (d) are the same in the range from -1 to 1, but are marked with different interval. It would be better to write them at the same interval. #4. There are many abbreviations in the text, so it would be better to define them when you first use them in the text. It would be nice to check it as a whole. #5. Table 3: In my opinion ‘AMRP’ should be modified ‘AMRD’. #6. Chapter 5: You have described the conclusions well, but I think it would be better to write them in more detail.Author Response
Response to the reviewer’ comments
We would like to thank the anonymous reviewer for reviewing our manuscript. We especially appreciate their constructive comments. We agree that some aspects of the paper needed further improvement and extension in order to make it more generally valid and suitable for publication. In the following, we address all comments point-by-point in more detail according to reviewers’ comments. All revisions are highlighted in the context.
Reviewer 2
Comments and Suggestions for Authors - Comments: At the first I would like to thank authors for very interesting work and hard work. This study examined temporal and spatial variations in the precipitation indices, and analyzed the relationships between the monthly precipitation and large-scale circulations in the upper and middle reaches of Heihe river basin, northwest China. The study also analyzed the relationship between precipitation and the NDVI and suggested that human activity could weaken the correlation between vegetation and natural precipitation. The results of this study will help to analyze the precipitation indices, and the relationship between precipitation and large-scale circulations/vegetation. Below are the comments, which I believe, can improve the work.
-Answer: We are very grateful for your positive evaluation and detailed comments.
Changes in precipitation can be central to quantifying the impact of climate change on the hydrological cycle. Also, changes in precipitation can mirror the variations of the hydrological cycle on a regional or global scale. Spatial and temporal variations of precipitation are, therefore, vital for determining the spatial and temporal distributions of water resources [12]. Therefore, the main objectives of this study were (1) to examine temporal and spatial variations in the precipitation indices, (2) to analyze the relationships between the monthly precipitation and large-scale circulations, and (3) to identify the relationship between precipitation and the NDVI. The results indicated that the occurrences (fractional contributions) of 1–2-day wet periods were 81.3% (93.9%) and 55.3% (82.1%) in the upper (UHRB) and middle (MHRB) reaches of the Heihe River Basin, respectively. The spatial distribution of the occurrences was almost consistent with non-significant increases/decreases at stations. The ATP, ATD, API, and AMRD increased, while precipitation regimes suggest that dry seasons were getting wetter, and wet seasons were getting drier, but these changes were not significant. Wavelet coherence analyses showed that climate indices influenced precipitation, mainly its concentration, on a 4- to 78.6-month timescale. The Pearson, Kendall, and Spearman correlation coefficients showed significant lagged correlations between precipitation and the North Arctic Oscillation (NAO), Pacific Decadal Oscillation (PDO), and Atlantic Multidecadal Oscillation (AMO). The NDVI was significantly and positively correlated with precipitation in the UHRB and significantly and negatively correlated with precipitation in the MHRB. The correlation between cultivated vegetation and natural precipitation in the MHRB may have been weakened by human activities.
#1. Table 2: Chapter 2 describes the study area and Table 2 does not seem to match the contents of this chapter. It may be necessary to move to another chapter or add a new chapter.
-Answer: Thanks for pointing this out. We revised it and move to section 4.5(P12 L332-333).
#2. Chapter 3.2.3: “Pearson, Kendall and Spearman tests” analyze correlations, so 'Correlation tests (Pearson, Kendall and Spearman)' would be better than 'Other Methods' for chapter titles (It would be nice if it would be modified to include all three). Also, it seems that there is a lack of explanation compared to the other methods above. It would be nice to add a description.
-Answer: Thanks for pointing this out. Considering the possible linear (Pearson) and non-linear (Spearman) correlations between precipitation and climate indices, we refer to De Oliveirajunior et al. (2018) who used these three methods for the detection of drought severity and its relationship to the ENSO and PDO climatic variability modes in the North and Northwest regions of the State of Rio de Janeiro, Brazil. The detailed calculation methods of three kinds of indices are also sourced from De Oliveirajunior et al. (2018). Here, according to Pearson, Kendall, and Spearman, we took the largest of two of correlation coefficients, and the significant levels were 0.05/0.01). We revised it and added the relevant information (P15 L400-419). For example:
……………………Here, according to Pearson, Kendall, and Spearman, we took the largest of two of correlation coefficients, and the significant levels were 0.05/0.01). Precipitation with a lag-time of between 0–12 months was not significantly correlated with ENSO, AO, and PNA in the UHRB (Table 4). However, the Pearson, Kendall, and Spearman’s tests showed that the NAO was significantly and positively correlated with precipitation at 0 month (no lag)……………..
#3. Figure 3: The color bars in the four figures (a) to (d) are the same in the range from -1 to 1, but are marked with different interval. It would be better to write them at the same interval.
-Answer: Thank you so much for pointing this out. We revised it in figure3. For example (P8 L257-258)
#4. There are many abbreviations in the text, so it would be better to define them when you first use them in the text. It would be nice to check it as a whole.
-Answer: Thank you so much for pointing this out. We have checked it as a whole. For example (P2 L49-50):
…………………. Recently, research has focused on the number of consecutive wet days (CWDs), known as wet periods (WPs) [11-12,16-18].
……………..
#5. Table 3: In my opinion ‘AMRP’ should be modified ‘AMRD’.
-Answer: Thank you so much for your comment. We revised it in the Table 2 (P11-12 L318).
#6. Chapter 5: You have described the conclusions well, but I think it would be better to write them in more detail.
-Answer: Thank you very much for your suggestion. We revised this part according to the comments (P18-19 L502-527). For example:
Spatial and temporal variations of WPs of occurrence and fractional contributions to the total precipitation amount (days), precipitation indices and the linkages with climatic indices were analyzed in the UHRB and MHRB. The precipitation episodes in the UHRB and MHRB were dominated by the occurrence and fractional contribution of WPs with durations of 1–4 days and 1–2 days, respectively. The occurrences and fractional contribution of longer-duration WPs decreased from the 1990s until 2016, while those of shorter WPs increased in the UHRB and MHRB. WPs occurred more frequently in the UHRB than in the MHRB, but the precipitation concentration was higher in the MHRB than in the UHRB. Thus, the risk of floods was higher in the UHRB than in the MHRB, while the drought risk may have grown in the MHRB. The spatial distribution of the fractional contributions and occurrences were almost consistent with the stations where the precipitation was increasing or decreasing.
The results of this analysis showed that the ATP, ATD, API, and AMRD increased in the UHRB and MHRB over the study period. Increases in the ATP and ATD caused an increase in the API in the UHRB and MHRB, whilst the AMRD also increased. The dry seasons became slightly wetter, and the wet seasons became slightly drier, but the precipitation regimes did not shift.
Precipitation and climate indices showed significant but variable correlations for ENSO, AO, NAO, PDO, PNA and AMO, especially for PDO, NAO and AMO at 4- to 78.6-month timescales. The results from Pearson, Kendall, and Spearman tests showed that there were significant lagged correlations between precipitation and the PDO (lag of 3 months), and AMO (lag of 1 month) indices in the UHRB, and NAO (lag of 12 months), PDO (lag of 3 months), and AMO (lag of 1 month) in the MHRB.
The NDVI increased significantly in the UHRB and MHRB from 1998 to 2015. It was significantly and positively correlated with precipitation in the UHRB and significantly and negatively correlated with precipitation in the MHRB. Human activities may have weakened the correlation between cultivated vegetation and natural precipitation in the MHRB.
These results point to new challenges for water supply and for water resources management under the impact of climate change and human activities, therefore, collaborative and sustainable water resources allocations are needed in the HRB.
References
[1] Gao: Y.; He, N.; Wang, Q.; Miao, C.Y. Increase of external nutrient input impact on carbon sinks in Chinese coastal seas. Environ. Sci. Technol. 2013,47,13215–13216. http://dx.doi.org/10.1021/es4045743.
[4] Yu, H.; Wang, L.; Yang, R.; Yang, M.L.; Gao, R. Temporal and spatial variation of precipitation in the Hengduan mountains region in China and its relationship with elevation and latitude. Atmos. Res. 2018,213,1-16. http://dx.doi.org/10.1016/j.atmosres.2018.05.025.
[5] Sayemuzzaman, M.; Jha, M.K. Seasonal and annual precipitation time series trend analysis in North Carolina, United States. Atmos. Res. 2014,137,183–194. http://dx.doi.org/10.1016/j.atmosres.2013.10.012.
[6] Yang, P.; Xia, J.; Zhang, Y.Y.; Hong, S. Temporal and spatial variations of precipitation in northwest china during 1960–2013. Atmos. Res. 2017,183,283-295. http://dx.doi.org/10.1016/j.atmosres.2016.09.014.
[11] Zhang, Q.; Singh, V.P.; Peng, J.; Chen, Y.D.; Li, J.F. Spatial–temporal changes of precipitation structure across the pearl river basin, China. J. Hydrol. 2012, 440-441(03),113-122. http://dx.doi.org/10.1016/j.jhydrol.2012.03.037.
[12] Zhang, Q.; Singh, V.P.; Li, J.F.; Chen, X.H. Analysis of the periods of maximum consecutive wet days in China. J. Geophys. Res Atmos. 2011, 116(D23),1-18. http://dx.doi.org/10.1029/2011JD016088.
[16] Zolina, O.; Simmer, C.; Gulev, S.; Kollet, K.S. Changing structure of European precipitation: Longer wet periods leading to more abundant rainfalls. Geophys. Res. Lett. 2010, 37, L06704. http://dx.doi.org/doi:10.1029/2010GL042468.
[17] Wang, Y.; Zhang, Q.; Singh, V.P. Spatiotemporal patterns of precipitation regimes in the Huai river basin, China, and possible relations with ENSO events. Nat. Hazards. 2016,82(3),1-19. http://dx.doi.org/10.1007/s11069-016-2303-3.
[18] Huang, J.; Chen, X.; Xue, Y.; Lin, J.; Zhang, J. Changing characteristics of wet/dry spells during 1961–2008 in Sichuan province, southwest China. Theor. Appl. Climatol. 2017,127(1-2), 129-141. http://dx.doi.org/10.1007/s00704-015-1621-9.
[19] Piao, S.L.; Fang, J.Y.; Ji,W.; Guo, Q.H.; Ke, J.H.; Tao, S. Variation in a satellite-based vegetation index in relation to climate in China. J. Veg. Sci. 2004,15,219–226. http://dx.doi.org/10.1658/1100-9233(2004)015[0219:VIASVI]2.0.CO;2.
[20] Cleland, E.E.; Chuine, I.; Menzel, A.; Mooney, H.A.; Schwartz, M.D. Shifting plant phenology in response to global change. Trends.Ecol. Evol. 2007, 22(7),357–365. http://dx.doi.org/10.1016/j.tree.2007.04.003.
[21] Mao, D.H.; Wang, Z.M.; Luo L.; Ren, C.Y. Integrating AVHRR and MODIS data to monitor NDVI changes and their relationships with climatic parameters in Northeast China. Int. J. Appl. Earth. Obs. Geoinf. 2012,18(1),528–536. http://dx.doi.org/10.1016/j.jag.2011.10.007.
[38] Li, B.F., Chen, Y.N., Chen, Z.S., Xiong, H.G., Lian, LS. Why does precipitation in northwest China show a significant increasing trend from 1960 to 2010. Atmos. Res. 2016, 275-284. http://dx.doi. org/10.1016/j.atmosres.2015.08.017.
[43] Qin, Y.H.; Li, B.F.; Chen, Z.S.; Chen, Y.N.; Lian, L.S. Spatio‐temporal variations of nonlinear trends of precipitation over an arid region of northwest China according to the extreme‐point symmetric mode decomposition method. Int. J. Climatol. 2017,38(11),1-11. http://dx.doi.org/10.1002/joc.5330
[66] Nalley, D.; Adamowski, J.; Khalil, B.; Biswas, A. Inter-annual to inter-decadal streamflow variability in Quebec and Ontario in relation to dominant largescale climate indices. J. Hydrol. 2016,536,426–446. http://dx.doi.org/10.1016/j.jhydrol.2016.02.049.
[67] De Oliveirajunior, J. F.; Gois, G.; Terassi, P. M.; Junior, C. A.; Blanco, C. J.; Sobral, B. S.; Gasparini, K. A. Drought severity based on the SPI index and its relation to the ENSO and PDO climatic variability modes in the regions North and Northwest of the State of Rio de Janeiro-Brazil. Atmos. Res. 2018, 212,91-105. http://dx.doi.org/10.1016/j.atmosres.2018.04.022.
Round 2
Reviewer 1 Report
Th authors revised the manuscript, but some issues haven’t been solved, had better check the data, table and figures to draw conclusions carefully.
1. In abstract, line 24-26. “The NDVI was significantly and positively correlated with precipitation in the UHRB and significantly and negatively correlated with precipitation in the MHRB”.
If check figure 1 c) and figure 11 b), the correlation vary with land cover type instead of UHRB/MHRB. Had better analyze it more carefully and revise the conclusion according to the figures.
2. In last review, I asked question about table 4 in Page 15. The most of the correlation coefficients are so small, and not statistically significant. So, can't draw meaningful conclusion based on these results. The authors cited [66], “Correlations are low, low correlation coefficients with significant p-values may occur due to the large sample size used in this study, which resulted in high degrees of freedom [66]”. But, it is about Spearman’s rank correlation. Table 4 lists 3 kind of correlations.
1) In table 4, Pearson correlation coefficients are really too low to get useful conclusion. Even p-value is less than 0.05, if the correlation coefficient is low, there is weak linkage between NDVI and precipitation.
2) In table 4, the correlation coefficients of Pearson, Kendall, and Spearman are quite different, even with different signs. How to explain this?

Author Response
We are very grateful for your evaluation and detailed comments. And we are revising this manuscript following your suggestions. I believe that it will lead to a great improvement in this manuscript.Please see the attachment.
